# Hydrophobic interactions dominate the recognition of a KRAS G12V neoantigen

Katharine M. Wright[1,2,3,16,19], Sarah R. DiNapoli [2,4,5,19],
Michelle S. Miller [1,2,3,17,19], P. Aitana Azurmendi[1,2,3], Xiaowei Zhao[6], Zhiheng Yu[6],
Mayukh Chakrabarti [1], WuXian Shi[7,8], Jacqueline Douglass[2,4,5],
Michael S. Hwang[2,4,5], Emily Han-Chung Hsiue[2,4,5,18], Brian J. Mog [2,4,5,9],
Alexander H. Pearlman [2,4,5], Suman Paul [4,5,10,11], Maximilian F. Konig [2,4,5,12],
Drew M. Pardoll[3,10], Chetan Bettegowda [4,5,10,13],
Nickolas Papadopoulos [4,5,10,14], Kenneth W. Kinzler[3,4,5,10],
Bert Vogelstein [2,3,4,5,10,14], Shibin Zhou [3,4,5,10] ✉ &
Sandra B. Gabelli [1,3,16,10,15] ✉

Specificity remains a major challenge to current therapeutic strategies for cancer. Mutation associated neoantigens (MANAs) are products of genetic alterations, making them highly specific therapeutic targets. MANAs are HLA-presented (pHLA) peptides derived from intracellular mutant proteins that are otherwise inaccessible to antibody-based therapeutics. Here, we describe the cryo-EM structure of an antibody-MANA pHLA complex. Specifically, we determine a TCR mimic (TCRm) antibody bound to its MANA target, the KRAS$^{G12V}$ peptide presented by HLA-A*03:01. Hydrophobic residues appear to account for the specificity of the mutant G12V residue. We also determine the structure of the wild-type G12 peptide bound to HLA-A*03:01, using X-ray crystallography. Based on these structures, we perform screens to validate the key residues required for peptide specificity. These experiments led us to a model for discrimination between the mutant and the wild-type peptides presented on HLA-A*03:01 based exclusively on hydrophobic interactions.

The *RAS* family (*HRAS, NRAS, KRAS*) of oncogenes is a highly sought-after target for cancer therapies[1]. More than 20% of several major cancer types, including pancreas, lung, and colon are driven by mutations in RAS genes, with *KRAS* comprising the bulk of those mutations[2–4]. The most common KRAS mutations occur at codon 12, where the wild-type glycine residue is mutated to a valine (G12V), cysteine (G12C) or aspartate (G12D) residue. For decades, targeting codon 12 mutant KRAS proteins with small molecule inhibitors was impeded by the inaccessibility of the GTP/GDP binding pocket, the featureless structure of KRAS, the lack of secondary binding pockets, and challenges in identifying selective inhibitors over the wild-type KRAS protein[5–10]. Recently, groups have identified covalent inhibitors to KRAS$^{G12C}$ and non-covalent inhibitors to KRAS$^{G12D}$ that offer promise for small molecule targeting of this previously "undruggable" target[10–17].

Immunotherapies targeting mutation-associated neoantigens (MANAs) are an alternative strategy to eliminate cancer cells harboring intracellular mutant oncoproteins, including KRAS. MANAs are processed and presented as short peptides on the cell surface, bound to a human leukocyte antigen (peptide-HLA complex, pHLA). These peptides are typically detected at single-digit to tens of copies per cell[18–20]. Multiple immunotherapy modalities can be used to target mutant pHLAs, including cancer vaccines[21,22], adoptive cell transfer of T cells expressing a pHLA-specific T cell receptor (TCR)[23–27], chimeric antigen receptors (CAR)[28], or bispecific antibodies[19,29].

KRAS[G12D] is presented on HLA-C*08:02 as the decamer G[10]A̲DGVGKSAL[19] (mutation underlined) or nonamer G[10]A̲DGVGKSA[18], and can be targeted with patient-derived TCRs[23,24,30]. Indeed, a patient receiving autologous T cells engineered to express two KRAS[G12D]-HLA-C*08:02-specific TCRs had a regression of her metastatic pancreatic cancer after a single infusion[31]. KRAS[G12V] is also processed and presented as a decamer V[7]VVGAV̲GVGK[16] (KRAS[G12V]$_{7-16}$) or nonamer V[8]VGAV̲GVGK[16] (KRAS[G12V]$_{8-16}$) on both HLA-A*03:01 and HLA-A*11:01[19,30,32,33]. Recently, we reported the generation of a bispecific antibody specific for the KRAS[G12V]$_{7-16}$ peptide presented on HLA-A*03:01. This bispecific antibody, named V2, is a single-chain diabody (scDb) that redirects T cells to kill cancer cells upon binding of one end of the scDb to CD3 in the TCR complex and binding of the other end to the KRAS[G12V] pHLA[19]. Despite the low antigen density of the KRAS[G12V] MANA on the surface of cancer cells, the V2 bispecific was highly sensitive and specific for the mutant pHLA over wild-type, and induced a robust T-cell response.

The determination of the structures of MANA-targeting therapeutics could yield unique information about how TCRs and antibody-based immunotherapies recognize pHLA, and provide opportunities for their improvement[28,34]. For example, structural analysis of the KRAS[G12D]-HLA-C*08:02 pHLA in complex with patient derived-TCRs revealed that the G12D mutation is a critical anchor residue for peptide presentation, but is not directly involved in TCR recognition of the neoantigenic peptide[23]. Others have compared affinity-enhanced TCRs and TCR mimic (TCRm) pHLA-targeting antibodies using crystal structures to understand the differences in binding affinity and specificity of agents with shared pHLA targets[28].

Here, we describe the structure of the V2 TCRm antibody against KRAS[G12V]$_{7-16}$-HLA-A*03:01, and demonstrate that hydrophobic interactions and an induced conformational change dictate V2 specificity for the KRAS[G12V] peptide.

## Results

### Single particle cryo-EM structure of V2-Fab bound to KRAS[G12V]-HLA-A*03:01

Initial attempts at structure determination were performed using the V2 single-chain variable fragment (scFv) in complex with the KRAS[G12V]-HLA-A*03:01 monomer. We could not get this protein expressed at high levels, despite several attempts, and so we switched to a full-length IgG format, grafting the V2 scFv into a full-length immunoglobulin G1 (IgG1) framework (V2-IgG). We performed pepsin-digestion and reduction of the V2-IgG into an antibody-fragment (V2-Fab') (Supplementary Fig. 1), but crystallization of the V2-Fab'-pHLA complex was unsuccessful. We then used size exclusion chromatography (SEC) to purify the full-length V2-IgG in complex with the KRAS[G12V]-HLA-A*03:01 monomer. Complex formation was confirmed by a shift in the SEC elution pattern (Fig. 1a, b). The high molecular weight of the V2-IgG/KRAS[G12V]-pHLA complex (150 kDa + 2 ×46 kDa) seemed ideal for single particle cryo-electron microscopy (cryo-EM).

After processing the collected cryo-EM micrographs (>6000 movies) on the V2-IgG/KRAS[G12V]-pHLA complex, initial blob particle picking and 2D classification didn't reveal the expected V2-IgG bound to two pHLA. The 2D classes appeared to belong to two distinct species: one arm of the pHLA-IgG complex, and an unbound IgG. We concluded that the hinge region of the IgG was too flexible to determine a full-length pHLA-IgG complex, so we concentrated on the V2-'Fab' portion bound to one KRAS[G12V]-pHLA monomer (Fig. 1c, -100 kDa structure). After template-picking particles that represented the Fab-pHLA complex (>3 million particles), subsequent two-dimensional (2D) and three-dimensional (3D) classifications were performed to give an initial overall reconstruction of the V2-Fab /KRAS[G12V]-HLA-A*03:01 complex to a resolution of 3.37 Å (Supplementary Figs. 2, 3). To improve the details of the map surrounding the binding interface, a final masked 3D classification which enveloped the V2-Fab-pHLA interface was performed. Subsequent refinement produced a final 3D reconstruction to an overall resolution of 3.14 Å (PDB ID 7STF) (Fig. 1d–f, Supplementary Fig. 2, Table 1). For manual rebuilding, we used previously determined structures of each component of the ternary complex with rigid body fitting. The final structure fits well into the map (Fig. 2a, b). Importantly, the region with the highest local resolution was the V2-Fab-pHLA interface, which consisted of the KRAS[G12V] peptide, the complementarity determining regions (CDRs) of the V2-Fab variable domains, and a region of the HLA-A*03:01 (the α1, α2 and β-sheet base) (Fig. 2c, d). By contrast, the constant domains of the V2-Fab showed the lowest local resolution (Fig. 2c, d). 3D variability analysis indicates high flexibility in the constant domains, and further highlights the challenge of obtaining projections of full-length V2-IgG due to the intrinsic flexibility of the hinge region (Supplementary Movie 1).

### Hydrophobic interactions dominate the interface between V2-Fab and KRAS[G12V]-HLA-A*03:01

The cryo-EM complex structure revealed that the V2-Fab sits on top of the KRAS[G12V]-HLA-A*03:01 binding groove in a partially parallel orientation, leaning heavily towards the C-terminus of the KRAS[G12V] peptide and α1 of HLA-A*03:01. The slope towards the C-terminus was consistent with previously reported functional data, where positional scanning mutagenesis along the KRAS[G12V] peptide showed that V2's specificity does not rely on the N-terminus of the peptide[19]. The calculated docking angle, and incident, or 'tilt', angles were 35° and 24°, respectively (Fig. 3a, b). The total buried surface area of the V2-Fab/KRAS[G12V]-HLA-A*03:01 was 1388 Å², with the variable domain of the heavy chain contributing to ~25% more than the variable light chain (862 Å² and 526 Å², respectively) (Table S1). Even though this surface area was smaller compared to other known TCRm- and TCR-pHLA structures (>2000 Å²)[28], other complexes achieve high specificity with even smaller footprints[18,35]. Importantly, the regions making up the binding interface were well-resolved in the EM map, specifically the KRAS[G12V] peptide, the CDRs of the V2-Fab, and the HLA-A*03:01 (Supplementary Fig. 4).

Overall, the V2-Fab made a total of 149 contacts with the HLA-A*03:01, mediated by five of the six CDRs, all three in the variable light chain, CDR[H1], and CDR[H2] of the heavy chain (Fig. 3c, Supplementary Table 1). Contacts were calculated with a 4 Å cutoff, which includes hydrogen bonds and van der Waals interactions. These contacts were a mixture of hydrogen bonds and hydrophobic interactions (Supplementary Fig. 5). By contrast, only three V2-Fab CDRs (CDR[H1,] CDR[H3] and CDR[L2]) interacted with the KRAS[G12V] peptide, making 26 contacts (14.9% of all contacts) (Fig. 3c, Supplementary Table 2), all of which are hydrophobic in nature (Supplementary Fig. 5). The majority of peptide contacts were centered between the C-terminus of the KRAS[G12V] peptide and the core structure of CDR[H3], which was composed of residues Asn101[H3], Ile102[H3], Pro103[H3], Val104[H3], and Tyr105[H3] (Fig. 3d, Supplementary Fig. 5). The V2-Fab engaged the site of the mutation, Val12, with a loose hydrophobic cage consisting of residues Pro103[H3], Val104[H3], Tyr105[H3], and Phe53[L2]. Specifically, Pro103[H3] sat in the small groove of the peptide, Val104[H3] sat directly on top for valine-valine contact, and Tyr105[H3] was positioned directly over the core of Val12[G12V], pointing towards α2 of HLA-A*03:01 (Fig. 3d, Supplementary Fig. 5). Phe53 from CDR[L2] was positioned laterally to Val104[H3], trapping Val12 in a hydrophobic environment. Ile102[H3] makes additional peptide contacts with the hydrophobic sidechains of Val14[G12V] and Gly15[G12V] (Fig. 3d). Moreover, residue Tyr31, located in CDR[L2], pointed down into the binding pocket, contacting the anchor residue Lys16[G12V] (Fig. 3d, Supplementary Fig. 5). Interestingly, the N-terminus of the KRAS[G12V] peptide was almost completely untouched by the V2-Fab.

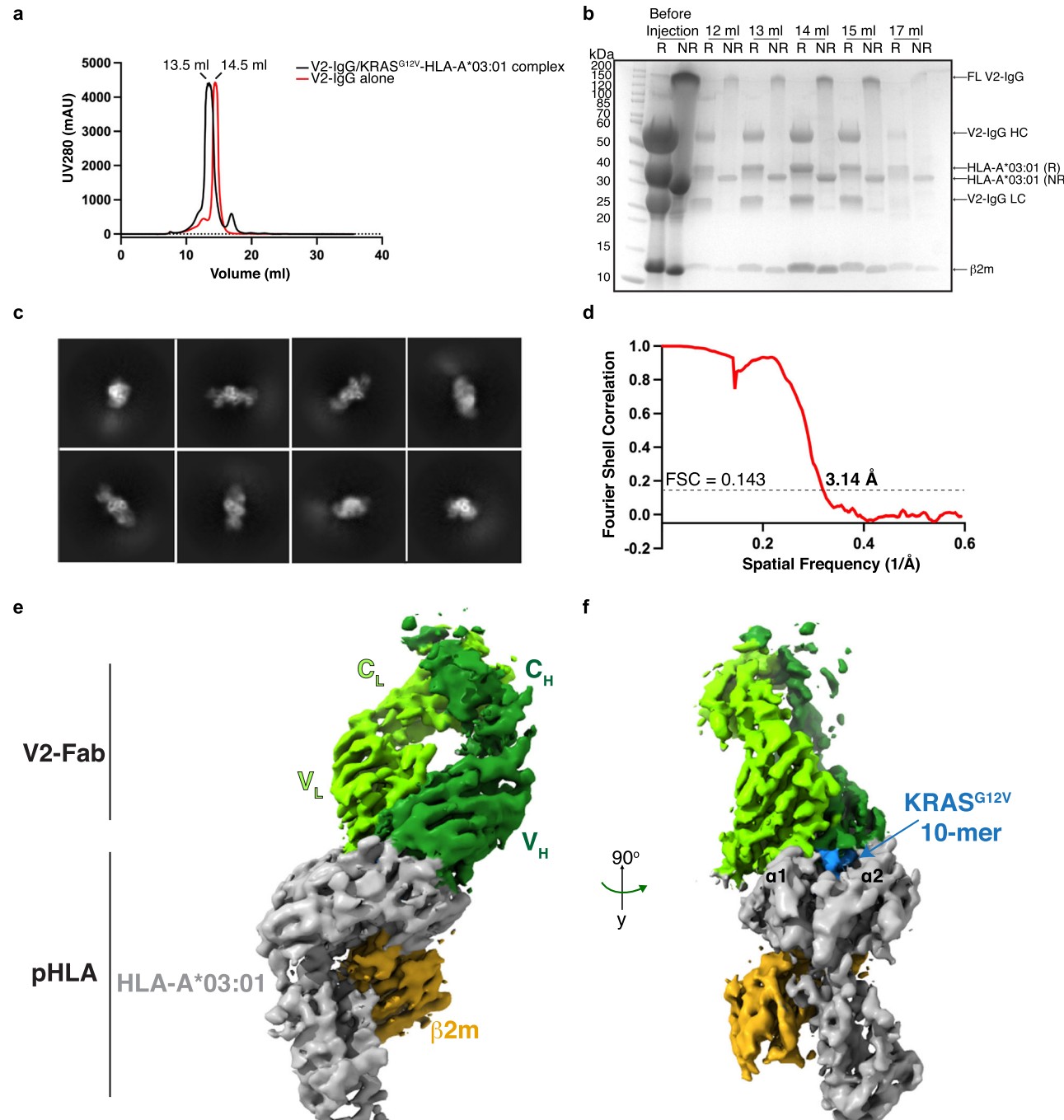

**Fig. 1 | 3D reconstruction of the V2-Fab/KRAS$^{G12V}$-HLA-A*03:01 complex. a** Size-exclusion chromatogram of the KRAS$^{G12V}$-HLA-A*03:01 in complex with the V2-IgG (black) overlayed with the chromatogram of the V2-IgG alone (red). Protein was monitored by absorbance at 280 nm, with an observed shift of 1 ml, proportional to the difference in molecular weight in the major peak. Retention volumes of the major peak are labeled. $N > 3$ independent experiments. **b** Coomassie-stained gradient SDS-PAGE gel of the eluted fractions at 12–17 ml from (**a**). R reducing 2× loading buffer, NR non reducing 2× loading buffer, HC heavy chain, LC light chain. $N > 3$ independent experiments. **c** Representative 2D classifications used for template picking. **d** The Fourier shell correlation (FSC) curve for the 3D reconstruction of the V2-Fab/KRAS$^{G12V}$-HLA-A*03:01 complex. With an FSC cutoff value of 0.143, the final resolution of the complex structure was 3.14 Å. **e** Cryo-EM density map of the V2-Fab/KRAS$^{G12V}$-HLA-A*03:01 complex at 3.14 Å. HLA-A*03:01 and β2 microglobulin (β2M) are colored in gray and gold, respectively. The V2-Fab is colored according to the heavy (dark green) and light (chartreuse) chains of the Fab fragment. **f** Cryo-EM map of V2-Fab/KRAS$^{G12V}$-HLA-A*03:01 at 90° to that shown in (**e**). The ten amino acid KRAS$^{G12V}$ peptide is shown in blue between helices α1 and α2 of the HLA-A*03:01.

## The KRAS$^{WT}$ peptide sits in the HLA-A*03:01 binding pocket

We determined the structure of the KRAS$^{WT}_{7-16}$ peptide bound to HLA-A*03:01, using X-ray crystallography to shed light on the specificity of the V2-Fab for KRAS$^{G12V}$. By means of molecular replacement, the crystal structure of KRAS$^{WT}$−HLA-A*03:01 was determined and refined to a resolution of 2.59 Å (PDB ID 8DVG) (Fig. 4a, b, Table 2). One pHLA

monomer was in the asymmetric unit, with well-resolved electron density observed for the KRAS$^{WT}$ peptide and the binding groove of HLA-A*03:01 (Supplementary Fig. 6a, b). The KRAS$^{WT}$ peptide was anchored into the peptide binding groove between the α1 and α2 helices by the preferred canonical anchor residues for HLA-A*03:01–a small hydrophobic residue, Val8$^{WT}$ (position 2), and a positively

**Table 1 | Cryo-EM data collection and refinement statistics**

| | V2-Fab–KRAS^G12V/HLA-A*03:01 (PDB ID 7STF) |
|---|---|
| **Data collection** | |
| EM equipment | Titan Krios |
| Voltage (kV) | 300 |
| Detector | Gatan K3 |
| Energy filter | Gatan Bioquantum |
| Pixel size (Å) | 0.844 |
| Electron dose (e⁻/Å²) | 60 |
| Defocus range (μm) | −0.8 to −2.0 |
| **Reconstruction** | |
| Software | RELION 3.1 |
| Number of used particles | 116,685 |
| Symmetry imposed | C1 |
| Overall resolution (Å) | 3.14 |
| **Refinement** | |
| Software | Phenix |
| Cell dimensions | |
| a, b, c (Å) | 70.05, 64.14, 145.17 |
| α, β, γ (°) | 90, 90, 90 |
| Model composition | |
| Protein residues | 817 |
| R.m.s. deviations | |
| Bonds (Å) | 0.011 |
| Angles (°) | 1.705 |
| Ramachandran (%) | |
| Favorable | 81.12 |
| Allowed | 17.76 |
| Outlier | 1.12 |
| Molprobity score | 3.46 |

charged residue, Lys16^WT (position 10)—with an overall buried surface area by the peptide of 743 Å² (Fig. 4a, b).

The majority of the amino acids making up the KRAS^WT peptide (V⁷VVGA**G**GVGK¹⁶) are small, hydrophobic residues, where the side-chains cannot make electrostatic or hydrogen bonding interactions. Therefore, most of the available hydrogen bonding interactions with the HLA-A*03:01 were with the backbone of the KRAS^WT peptide. In total, 10 hydrogen bonds and one electrostatic interaction were made between the KRAS^WT peptide and HLA-A*03:01 (Fig. 4b, Supplementary Fig. 6c). Specifically, the N-terminus Val7^WT is anchored by three tyrosine residues of HLA-A*03:01: two on the α2 (Tyr159, Tyr171), and one on the β-sheet base (Tyr7) (Fig. 4b). The backbone amide of the anchor residue Val8^WT was stabilized by a hydrogen bond with Glu63 of α1 of HLA-A*03:01. The side chains of residues Val9^WT and Gly10^WT undergo hydrophobic interactions with HLA-A*03:01 residues, while the amide of Val9^WT made a hydrogen bond with Tyr99 on the β-sheet base (Fig. 4b). The main chain of Ala11^WT was stabilized by a hydrogen bond to HLA-A*03:01 Asn66 on α1. Gly12^WT, Gly13^WT, and Val14^WT were stabilized by multiple hydrophobic residues in HLA-A*03:01, with no direct hydrogen bonds to the backbone of these residues. The site of the mutation, Gly12^WT (position 6 in the peptide), was located in the center of the peptide, positioned outside of the HLA-A*03:01 binding groove (Fig. 4b). The carbonyl of Gly15^WT was secured by Tyr147 of α2. Lastly, the C-terminus of the peptide was heavily anchored into HLA-A*03:01. The carboxyl group of Lys16^WT was secured by Tyr84 (α1) and Thr143 (α2), while the amino group was near Asp77 on α1. The side chain of Lys16^WT, which is the only peptide side chain capable of making hydrogen bond interactions, formed an electrostatic interaction with Asp116 of the β-sheet base of HLA-A*03:01 (Fig. 4b).

For a direct comparison of the position of the KRAS^G12V peptide within HLA-A*03:01, we extensively attempted to crystallize the KRAS^G12V–pHLA monomer alone but were unsuccessful. Therefore, we analyzed the interactions between the mutant peptide and HLA using the cryo-EM complex structure. The KRAS^G12V peptide bound to HLA-A*03:01 had a buried surface area of 730 Å², which is 13 Å² less than the KRAS^WT peptide. A total of 4 hydrogen bonds and one electrostatic interaction were made between the KRAS^G12V peptide and HLA-A*03:01 compared to 10 and 1, respectively, in the KRAS^WT-HLA-A*03:01 (Fig. 4c, d). Val7^G12V was anchored by the same three tyrosine residues observed with KRAS^WT (Tyr7, Tyr159, Tyr171). Whereas residues Val8^WT, Val9^WT and Ala11^WT formed hydrogen bonds to HLA-A*03:01, residues Val8^G12V-Gly13^G12V were only involved in hydrophobic interactions, with no direct stabilization of the backbone. The site of mutation, Val12^G12V, was pointing up, toward α1 in the HLA-A*03:01 binding groove (Fig. 4d, Supplementary Fig. 6d). The backbone amine of Val14^G12V was stabilized by a hydrogen bond with the side chain of Glu152 of α2 of HLA-A*03:01, an interaction not observed in the KRAS^WT structure. The Lys16^G12V side chain had a similar electrostatic interaction with Asp116 to Lys16^WT, but its C terminus was not anchored by multiple residues on α1 and α2 of HLA-A*03:01 as in the KRAS^WT structure. The fewer interactions made between the KRAS^G12V peptide and HLA-A*03:01 correlates with the slightly lower buried surface area.

To rule out the possibility that crystallization artifacts contributed to the conformation of the KRAS^WT peptide bound to HLA-A*03:01, we conducted molecular dynamics simulations of this system, amounting to an aggregate sampling time of 1.02 μs. In these simulations, the distance between the complex and the edge of the solvent box was 1.2 nm, such that any periodic images of the KRAS^WT-pHLA complex were 2.4 nm apart. The root mean-squared fluctuation (RMSF) data for the residues of the KRAS^WT peptide over the course of this sampling time, a metric that quantifies the standard deviation of the residues about their time-averaged position, was less than 1.5 Å for all peptide residues except for Gly7^WT (Supplementary Fig. 7). This observation indicates that the KRAS^WT peptide remains stably associated with the HLA-A*03:01 monomer and crystal contacts have not contributed significantly to the observed binding conformation.

**Induced fit of KRAS^G12V-pHLA by the V2-Fab underlies specificity**

Considering the hydrophobic nature of the V2-Fab/KRAS^G12V-pHLA interaction, the determinants of specificity were still not completely understood. Therefore, in the absence of the KRAS^G12V–pHLA monomer structure, the thermal stability of each monomer was evaluated using differential scanning fluorimetry (DSF) to assess the role of the G12V mutation on the observed changes in pHLA interactions. Both the mutant and WT pHLA had a similar melting temperature around 53–54 °C (Fig. 5a). The comparable T_m suggested that the pHLA monomers had a similar structure.

Given the stability measurements and suggested similarity in the pHLA structures, we sought to better comprehend the structural basis of recognition by comparing the unbound KRAS^WT pHLA structure with the V2-Fab bound KRAS^G12V structure. We performed a structural alignment of the HLA-A*03:01 from the KRAS^G12V-bound and KRAS^WT-unbound structures, and as expected for a structural alignment with a rmsd of 1.014 Å over 275 Cα, the HLA-A*03:01 molecules did not show major conformational changes, especially within the peptide binding groove (Fig. 5b). There was very slight movement of the C-terminal end of the α2. The major conformational change observed was in the KRAS peptide, where KRAS^G12V was more N-terminally shifted compared with the KRAS^WT peptide within the binding groove (Fig. 5c). There were only three residues within the KRAS^G12V peptide that maintained the same conformation as KRAS^WT (Val7, Val8, Val9 with an rmsd of 0.4-0.5 Å). The other 7 amino acids underwent a backbone displacement.

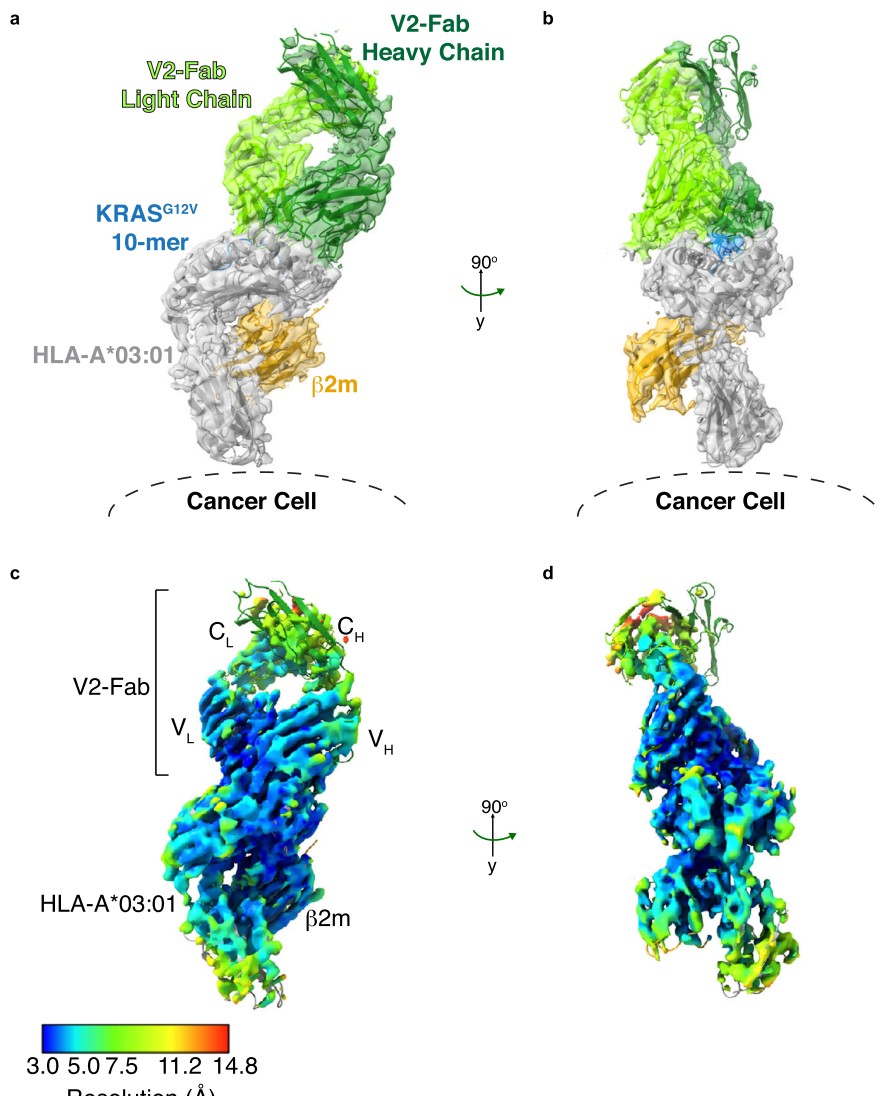

**Fig. 2 | Structure of V2-Fab/KRAS^G12V-HLA-A*03:01. a** The final rebuilt model of V2-Fab/KRAS^G12V-HLA-A*03:01 (PDB ID 7STF), with the cryo-EM map superimposed and in a transparent view. The structure and map are colored according to Fig. 1e. To aid the conceptual visualization, the location of the HLA-A*03:01 relative to the surface of the cancer cell is highlighted. **b** Structure and density map of V2-Fab/ KRAS^G12V-HLA-A*03:01 at 90° to the view shown in (**a**). **c** Local resolution map of V2-Fab/KRAS^G12V-HLA-A*03:01 in the same orientation as (**a**). Resolution map was calculated in MonoRes and depicted in ChimeraX. **d** Local resolution map of V2-Fab/ KRAS^G12V-HLA-A*03:01 at 90° to that shown in (**c**).

Residues Gly10, Ala11 and Val12 of the KRAS^G12V peptide had some of the largest observed shifts compared to KRAS^WT (rmsd 1.14, 2.78 and 1.86 Å, respectively, as measured at their Cα) (Fig. 5c). Moreover, the binding of the V2-Fab had pushed the C-terminus of the peptide, and particularly residue Lys16, deeper into the binding pocket (rmsd 2.13 Å) (Fig. 5c). This buried C-terminus further pushed the other peptide residues, Gly10 and Ala11, toward the N-terminus of the binding groove, resulting in this overall induced fit (Fig. 5b, c). Taken together with the V2-Fab/KRAS^G12V-pHLA interactions, we hypothesize that the induced fit observed upon V2-Fab binding was driven by the loose, hydrophobic cage formation made between the CDRs and residue Val12 in KRAS^G12V-pHLA.

To further corroborate our hypothesis that the V2-Fab induces a conformational change in the KRAS^G12V peptide via formation of a hydrophobic cage, we conducted molecular dynamics simulations of KRAS^G12V-pHLA and a system in which the original KRAS^G12V-pHLA structure was reverted to KRAS^WT-pHLA prior to simulation. Both simulations were performed in the absence of the V2-Fab. As such, these simulations explored the effect of having the KRAS^G12V peptide, previously in contact with the V2-Fab, now exposed to solvent. Simulations were conducted for an aggregate sampling time of 1.02 μs for both KRAS^G12V-pHLA and the system in which KRAS^G12V-pHLA was reverted to KRAS^WT. The RMSF plots for KRAS^G12V and KRAS^G12V reverted to KRAS^WT both indicate larger fluctuations for all peptide residues relative to the fluctuations observed for the original KRAS^WT system (Supplementary Fig. 7). Importantly, the peptide still maintained association with HLA-A*03:01 throughout the course of the simulations. The fluctuations observed for the KRAS^G12V system reverted to KRAS^WT, intermediate to both the KRAS^G12V and the original KRAS^WT, may suggest a "memory" effect originating from the original conformation in which KRAS^G12V was interacting with the V2-Fab, tempered by its reversion to KRAS^WT prior to simulation. This result has two implications: (1) it suggests that the absence of the V2-Fab, and thereby the loss of previously stabilizing hydrophobic interactions, results in an increase in KRAS^G12V peptide dynamics, and (2) that the increased dynamics of the KRAS^G12V peptide over the original KRAS^WT may contribute to its ability to adopt conformations that enhance the favorability of the

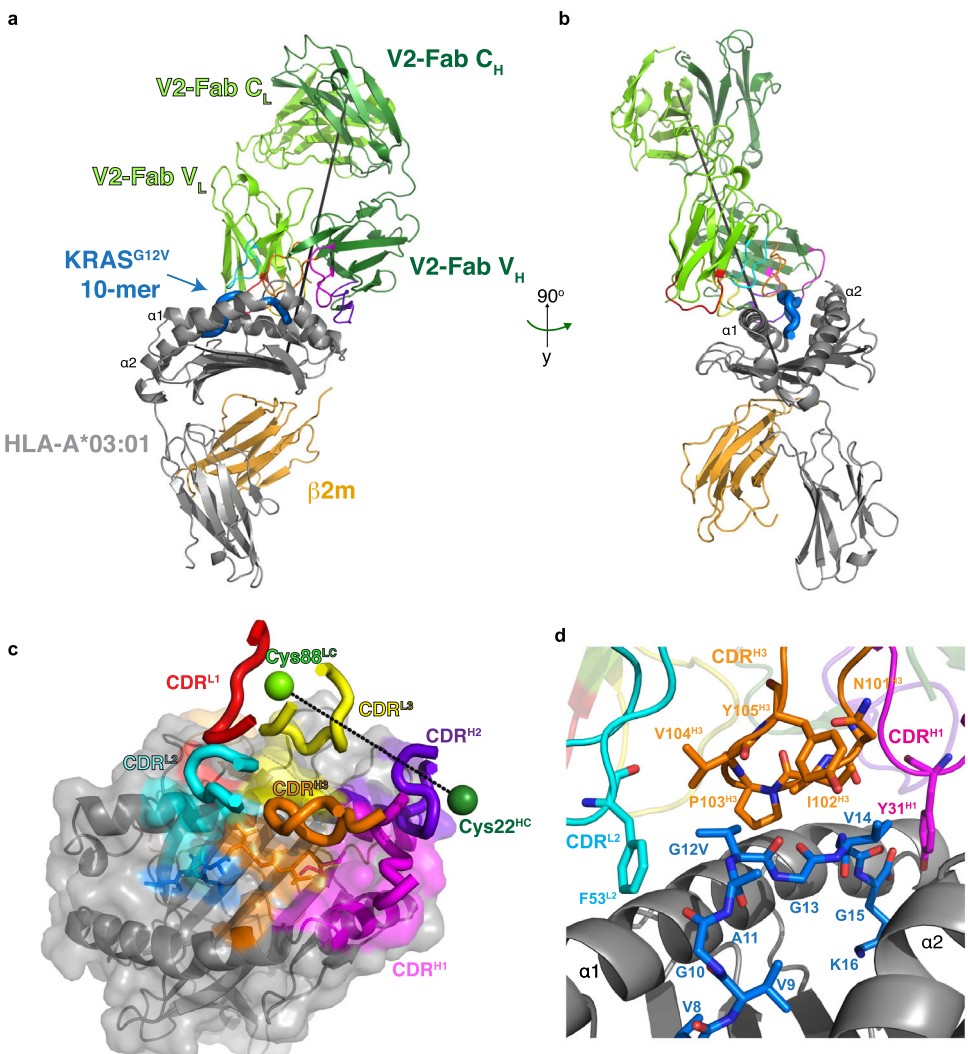

**Fig. 3 | Structural basis of recognition of V2-Fab. a** Overall structure of V2-Fab/KRAS$^{G12V}$-HLA-A*03:01 (PDB ID 7STF). The structure is colored according to Fig. 2. The CDRs are colored: L1 (red), L2 (cyan), L3 (yellow), H1 (magenta), H2 (purple), H3 (orange). The incident or 'tilt' angle is represented by a black line. **b** Structure of the V2-Fab/KRAS$^{G12V}$-HLA-A*03:01 complex shown in (**a**) at 90° rotation of that view. **c** Bird's-eye view of the surface representation of the HLA-A*03:01 shown in gray, KRAS$^{G12V}$ peptide shown in blue, and the contacting residues colored according to labeled CDRs of the V2-Fab, as in (**a**). The crossing angle, calculated from a vector between Cys22 from the V2 variable heavy chain and Cys88 from the V2 variable light chain, is shown as a dashed line. **d** Zoomed-in view of the KRAS$^{G12V}$ peptide (aa Val8-Lys16) centered on the site of mutation, with interacting residues of CDR$^{H3}$ (orange) and CDR$^{L2}$ (cyan) shown as sticks.

induced conformational change upon V2-Fab interaction, contributing to recognition specificity.

**Affinity maturation of V2**

Bispecific antibody affinity has been positively correlated with potency across a variety of targets[20,28,36–40]. In particular, targeting low antigen density molecules, such as pHLA complexes, may require low nanomolar to picomolar range affinities for therapeutic use[20]. To determine whether we could improve the potency of the V2 scDb by increasing its affinity from its previously reported $K_D$ of 24 nM[19], we designed and synthesized a library of single amino-acid variants consisting of 61 sites across the 6 CDRs, with each of the other 19 amino acids represented at each site, for a total of 1159 variants. Library diversity was characterized by next generation sequencing (Supplementary Fig. 8, Supplementary Data 1). Three variants were not detected by sequencing: A32M (CDR$^{L1}$), V50M, and D54W (both CDR$^{H2}$). The overall average variant frequency per site was $5.092 \pm 1.394\%$. V2 variants specific for the KRAS$^{G12V}$–pHLA monomer were enriched over five rounds of negative and positive selection using the variant phage display library. After

rounds 4 and 5 of panning, individual phage clones were selected for characterization. Sanger sequencing revealed a high diversity after rounds 4 and 5, with variants present across the six CDRs (Supplementary Fig. 9a), and 96% of variants were represented by only 1 or 2 colonies (Supplementary Fig. 9b). To determine the relative binding of each variant to KRAS$^{G12V}$–pHLA or KRAS$^{G12WT}$–pHLA monomer, phage clones were screened by ELISA. With the exception of F53T in CDR$^{L2}$ and G56K in CDR$^{H2}$, all variants retained specificity for the mutant KRAS$^{G12V}$ monomer over the KRAS$^{WT}$ monomer (Supplementary Fig. 9c).

Though most variants retained specificity for the KRAS$^{G12V}$-pHLA monomer, the high overall diversity among the variants and distribution across the CDRs gave little indication of which variants to select for functional characterization as scDbs. We therefore chose variants using structural information. Specifically, we aimed to introduce new types of interactions between V2 and KRAS$^{G12V}$–pHLA to supplement existing hydrophobic interactions. We focused on V2 residues that formed a loose cage around KRAS Val12, including Ile102, Val104, Tyr105 in CDR$^{H3}$, and Phe53 in CDR$^{L2}$. While variants for Tyr105$^{H3}$ were

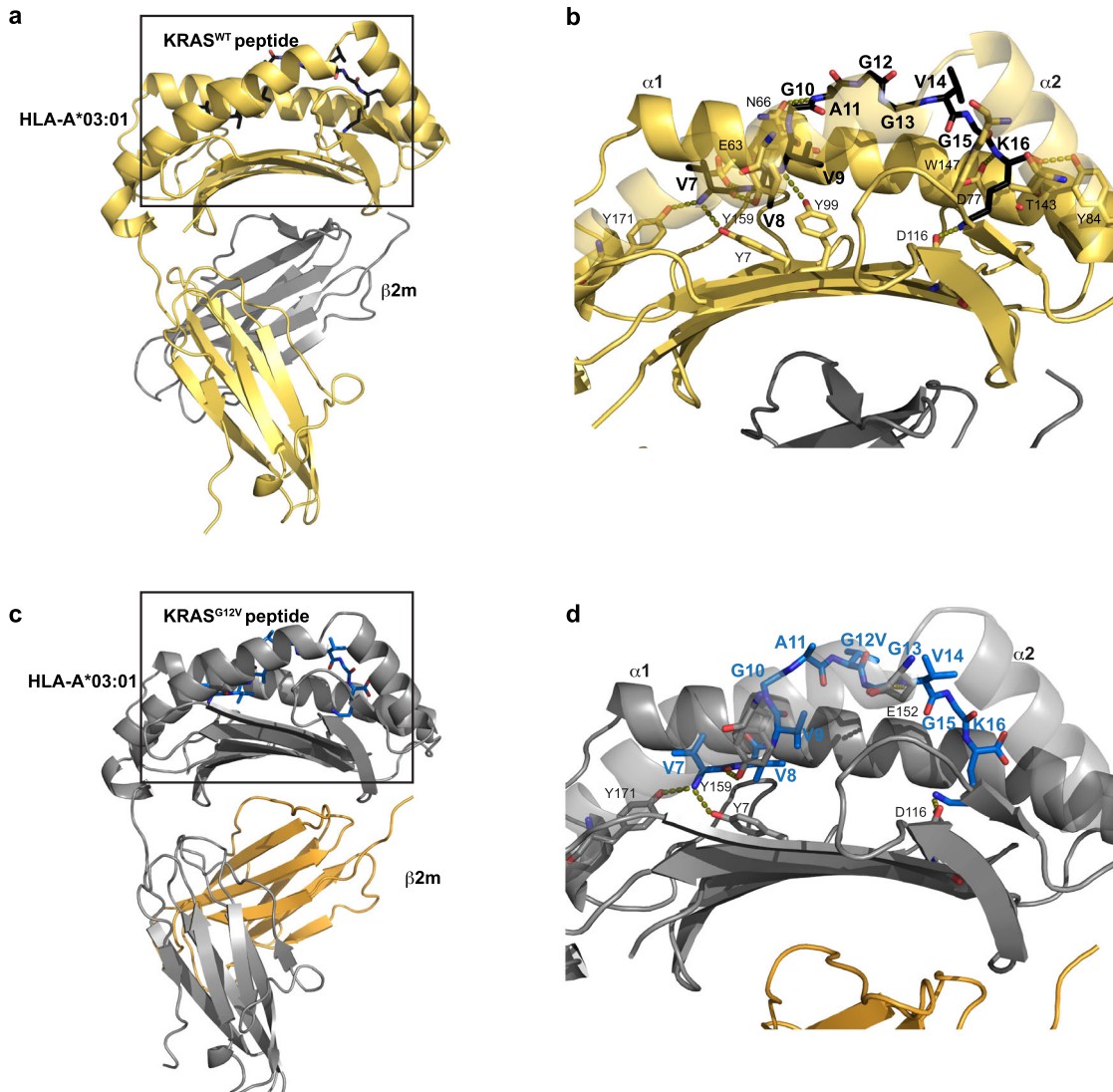

**Fig. 4 | KRAS^G12V mutation is protruding out of the HLA binding groove.**
**a** Overall X-ray structure of the KRAS^WT peptide bound to HLA-A*03:01 (PDB ID 8DVG). β2 microglobulin (β2M) and HLA-A*03:01 are colored in gray and gold, respectively. The 10-mer KRAS_7–16 peptide is colored in black. The box highlights the zoomed in region shown in (**b**). **b** Detailed interactions of the KRAS^WT peptide with HLA-A*03:01. The side chain of the interacting residues of HLA-A*03:01 (yellow) and peptide (black) are shown as sticks. Hydrogen bonds are represented as dashed lines. **c** Structure of the KRAS^G12V peptide bound to HLA-A*03:01, extracted from the cryo-EM complex structure. The HLA-A*03:01, β2M, and KRAS^G12V peptide are colored according to Fig. 2a. The box highlights the zoomed in region shown in (**d**). **d** Detailed interactions of the KRAS^G12V peptide with HLA-A*03:01. The side chain of the interacting residues of HLA-A*03:01 (gray) and peptide (blue) are shown as sticks. Hydrogen bonds are represented as dashed lines.

not present in the colonies sampled after phage panning, we decided to mutate this residue due to its location above the KRAS^G12V peptide. We were particularly interested in the overall role of Tyr105^H3 in binding, since it makes a hydrogen bond with Gln155 on α2 of HLA-A*03:01, crossing over the peptide. Variants for Pro103^H3 were also not present after panning, and further mutations were not generated due to its suspected importance to CDR^H3 loop conformation. In addition, Ala106 in CDR^H3 and Gln89 in CDR^L3 were included, based on their abundance in the monoclonal phage selected after panning. A total of 18 single amino acid variants of V2 were generated by site-directed mutagenesis (Supplementary Table 3). V2 and its variants were expressed as scDbs in small-scale HEK293FT cultures and purified by Ni-NTA resin-based His purification for functional testing.

## V2 variants with increased affinity to KRAS^G12V peptide
To estimate the V2 variants' binding affinity relative to the original V2 scDb, scDbs were incubated on an ELISA plate coated with CD3ε/

CD3δ heterodimer, the KRAS^G12V–pHLA, or the KRAS^G12WT–pHLA monomer (Fig. 6a, Supplementary Fig. 10). All Tyr105^H3 variants displayed a loss of binding on ELISA (Fig. 6a). The Phe53^L2 variants, along with V104I^H3 and V104N^H3, also had lower relative binding compared with the original V2 scDb. Q89D^L3, I102T^H3, V104A^H3, V104R^H3, A106I^H3, A106L^H3, A106M^H3, and A106T^H3 all had higher relative binding to the KRAS^G12V–pHLA monomer compared to the original V2 scDb. No variant had more than minimal binding to the KRAS^WT–pHLA monomer (Supplementary Fig. 10).

We further employed surface plasmon resonance (SPR) to measure the binding kinetics and affinity of three V2 scDb variants. We chose F53W^L2, V104N^H3, and V104R^H3 to compare to the original V2 scDb, due to the varying relative binding observed (Fig. 6a), and their essential role in the interaction with the KRAS^G12V peptide. SPR analysis of the original V2-scDb revealed a $K_D$ of 34 nM to KRAS^G12V–pHLA (Fig. 6b, Supplementary Table 4), corresponding well with published data. F53W^L2 had lower relative binding on ELISA, but

**Table 2 | Crystallization data collection and refinement statistics**

| | KRAS$^{WT}$/HLA-A*03:01 (PDB ID 8DVG) |
|---|---|
| **Data collection** | |
| Diffraction source | NSLS-II X17-ID-2 |
| Wavelength (Å) | 0.979321 |
| Temperature (K) | 100 |
| Detector | Dectris EIGER X 16 M |
| Space group | P622 |
| *a, b, c* (Å) | 152.9, 152.9, 85.2 |
| α, β, γ (°) | 90, 90, 120 |
| Resolution range (Å) | 19.65–2.59 (2.69-2.59) |
| Total no. of reflections | 192,744 (18,422) |
| No. of unique reflections | 18,642 (1,809) |
| Completeness (%) | 99.6 (99.1) |
| Redundancy | 10.3 (10.1) |
| $\langle I/\sigma(I)\rangle$ | 9.0 (1.9) |
| $R_{merge}$ | 0.240 (1.18) |
| $R_{meas}$ [a] | 0.252 (1.24) |
| $R_{pim}$ [b] | 0.077 (0.384) |
| $CC_{1/2}$ | 0.99 (0.74) |
| **Refinement** | |
| Resolution range (Å) | 19.65–2.59 (2.66–2.59) |
| No. of reflections, working set | 17,735 |
| $R_{work}/R_{free}$ | 0.191/0.0.228 (0.250/0.285) |
| Total atoms | |
| Protein | 3153 |
| Water molecules | 112 |
| Ligands | 72 |
| R.m.s. deviations | |
| Bonds (Å) | 0.002 |
| Angles (°) | 0.517 |
| Average B factors (Å$^2$) | |
| Protein | 42.3 |
| Water molecules | 42.8 |
| Ligands | 101.6 |
| Ramachandran (%) | |
| Favorable | 97.9 |
| Outlier | 0 |

[a]$R_{meas}$ = Sum(Sqrt(N/(N-1))(|Ihl − <Ih>|)/Sum(<Ih>).

[b]$R_{pim}$ = Sum(Sqrt (1/N-1))(|Ihl − <Ih>|)/Sum(<Ih>).

a higher affinity by SPR (K$_D$ = 10.6 nM), indicating that affinity measurement by SPR may be more sensitive to smaller changes in affinity than ELISA (Fig. 6c, Supplementary Table 4). Similarly to the relative binding on ELISA, V104R$^{H3}$ scDb bound more tightly to KRAS$^{G12V}$–pHLA (K$_D$ = 6.6 nM), while V104N$^{H3}$ bound more weakly, with a K$_D$ = 185.5 nM compared with original V2 scDb (Fig. 6d, e; Supplementary Table 4). All scDbs had some binding to KRAS$^{WT}$–pHLA at the highest concentration, with V104R$^{H3}$ displaying the highest response (Fig. 6b–e). All SPR data were best fit to a two-state binding model, suggesting a conformational change upon binding, which aligns with the structural data. Importantly, the mutations did not result in protein unfolding, highlighted by the same stable melting temperature observed in the differential scanning fluorimetry assays (Supplementary Fig. 11).

To determine whether variants with increased binding in vitro could detect KRAS$^{G12V}$$_{7–16}$ peptide displayed on the cell surface, human T cells were co-cultured with T2A3 cells pulsed with either the KRAS$^{G12V}$$_{7–16}$ peptide or KRAS$^{WT}$$_{7–16}$ peptide (Fig. 7). Though some variants demonstrated stronger binding compared to the original V2 scDb, no variant displayed increased sensitivity to low peptide concentrations in this assay. Interestingly, the different variants of Val104$^{H3}$–which sits atop KRAS Val12– had different functional profiles. The V104N$^{H3}$ variant showed a reduced ability to induce IFNγ secretion compared to V2 at the same peptide concentration, in line with its lower binding affinity. This result is consistent with other reports where lower affinity bispecific antibodies have reduced ability to stimulate IFNγ release[20]. However, despite its increased measured affinity, V104R$^{H3}$ scDb did not significantly improve function compared to V2. Notably, all variations to the Tyr105$^{H3}$ residue resulted in a total loss of peptide-dependent T-cell activation, in line with loss of binding as shown by ELISA and lack of detection of Tyr105$^{H3}$ variants after panning.

## Comparing V2 variant activity against KRAS$^{G12V}$ at endogenous levels

For further analysis, we selected V2 variants with[1] comparable sensitivity and specificity to the original V2 scDb in the peptide-pulsing co-culture assay, and[2] at least 50% of V2 scDb's relative binding to the KRAS$^{G12V}$–pHLA monomer. These variants were co-cultured with an isogenic pair of NCI-H358-derived cell lines: a parental clone harboring the KRAS$^{G12C}$ mutation, and a clone with the KRAS$^{G12V}$ mutation knocked in to its endogenous locus[19]. scDbs at varying concentrations were incubated with T cells from two human donors, and the G12C- or G12V-containing NCI-H358 isogenic cell lines, to assess their ability to drive G12V-dependent cytotoxicity and IFNγ release (Fig. 8a). Compared to the original V2 scDb, none of the 8 variants had both retained specificity and increased T-cell activation, as measured by IFNγ release (Fig. 8b) and cytotoxicity (Fig. 8c). Several variants with higher IFNγ release than V2 scDb (Q89D$^{L3}$, A106L$^{H3}$, A106M$^{H3}$), along with A106T$^{H3}$, showed increased off-target activity against the G12C parental cell line, particularly the A106$^{H3}$ variants. The increase in H358$^{G12C}$ cytotoxicity may reflect that A106$^{H3}$ variants exhibit shared reactivity to both KRAS$^{G12V}$ and KRAS$^{G12C}$ but could be explained by cross-reactivity against other peptides presented on the H358 cell line. However, the A106$^{H3}$ variants also had an elevated baseline IFNγ secretion profile in the T2A3 co-culture assay that did not increase with higher concentrations of KRAS$^{WT}$ peptide. F53W$^{L2}$, V104A$^{H3}$, and V104I$^{H3}$ variants did not have improved activity or specificity over the V2 scDb.

## V2 scDb and its variants are not cross-reactive to KRAS$^{G12V}$ presented by HLA-A*11:01

Previously, we assessed cross reactivity of the V2 scDb to other HLA-A*03:01-binding peptides[19]. While cross reactivity with unrelated peptides poses a safety risk for MANA targeting, shared reactivity to the same peptide on other related HLAs could expand the population of patients who could benefit from the same therapeutic. In the United States, the 10 most common HLA alleles range in frequency from 16-42% of the population, with HLA-A*03:01 present at a frequency of 22%[1]. To potentially reach more patients with a KRAS$^{G12V}$ mutation, one strategy could be targeting the mutant KRAS peptide presented on HLA-A3 superfamily members. The KRAS$^{G12V}$$_{7–16}$ decamer can also be presented with high affinity by HLA-A*11:01, an HLA-A3 superfamily member[30,32,33]. HLA-A*11:01 is one of the most frequent class I HLA molecules in southeast Asia, with frequencies up to 40%[41]. Sequence alignment of HLA-A*11:01 (PDB ID 5WJN[42]) and HLA-A*03:01 revealed 97% sequence homology, with five positions of nonconserved changes and two of conserved changes in amino acid sequence (Supplementary Fig. 12a). Mapping of these seven different residues on the cryo-EM

**a**

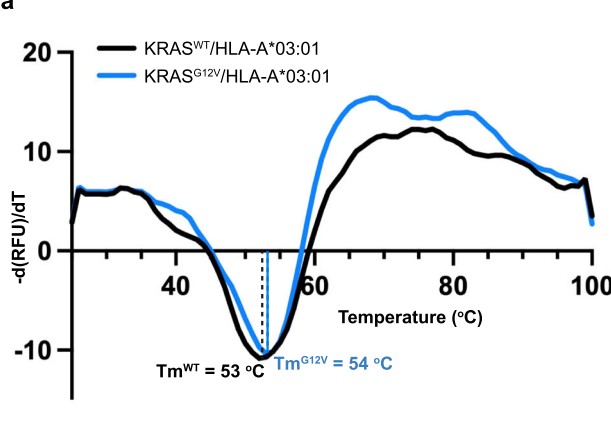

**b**

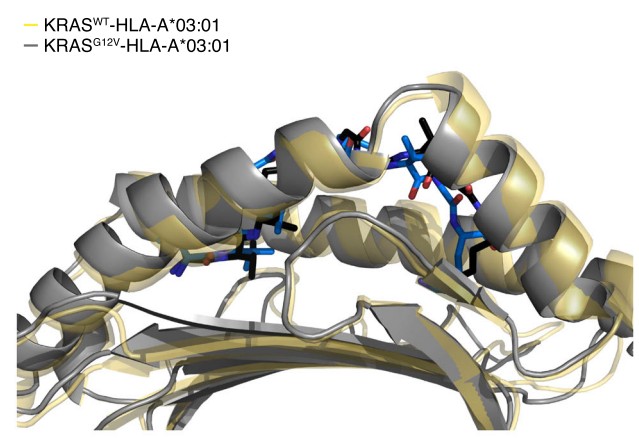

**c**

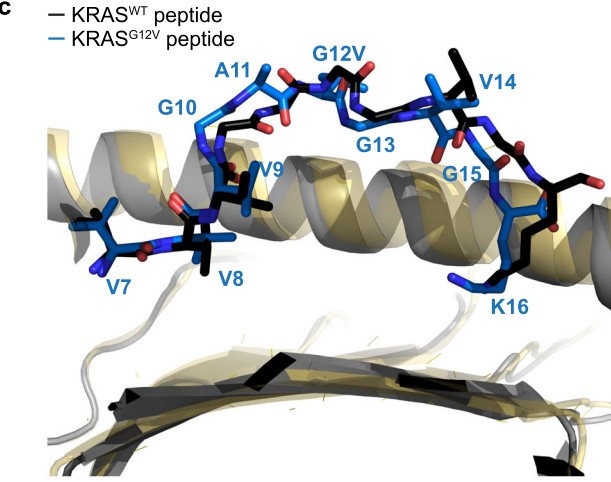

**Fig. 5 | Peptide conformational change influences specificity. a** Differential scanning fluorimetry analysis of KRAS$^{WT}$-HLA-A*03:01 (black) and KRAS$^{G12V}$-HLA-A*03:01 (blue). The negative derivative of relative fluorescence unit vs temperature is shown for each. The melting temperatures are labeled and correspond to the minimum peak of each first derivative. **b** Structural alignment of the KRAS$^{WT}$-HLA-A*03:01 (peptide, black sticks; HLA, yellow ribbons) and V2-Fab/KRAS$^{G12V}$-HLA-A*03:01 (peptide, blue sticks; HLA, gray ribbons) structures, highlighting the peptide binding groove. **c** The same structural alignment as in (**b**), with further zoom-in highlighting the conformational change within the KRAS$^{G12V}$ peptide upon V2-Fab binding.

V2-Fab-KRAS$^{G12V}$-HLA-A*03:01 structure revealed that they are not located at the binding interface between the antibody and pHLA. However, two residues were located within the pHLA binding groove (Glu152 and Thr163) (Supplementary Fig. 12b). Of particular interest was Glu152, which made a hydrogen bond with the backbone of Val14 on the KRAS$^{G12V}$ peptide, but not the KRAS$^{WT}$ peptide (Supplementary Fig. 6c, d).

Using SPR, we measured the binding kinetics of V2 and three scDb variants (F53W$^{L2}$, V104N$^{H3}$, and V104R$^{H3}$) to KRAS$^{G12V}_{7-16}$-HLA-A*11:01. SPR analysis revealed that despite the high sequence identity between HLA-A*11:01 and HLA-A*03:01, there was minimal binding to KRAS$^{G12V}_{7-16}$-HLA-A*11:01 (Supplementary Fig. 12C, Supplementary Table 4). When fit to a two-state model, original V2 scDb and V104R$^{H3}$ had calculated K$_D$ values > 3.2 μM (maximum concentration) (Supplementary Table 4). This was a > 400-fold reduction in affinity with KRAS$^{G12V}$ bound to HLA-A*11:01 for original V2 scDb, and over 1000-fold reduction for V104R$^{H3}$ scDb. There was no appreciable binding to KRAS$^{WT}$-HLA-A*11:01 either. Interestingly, the response levels were equal to that observed when the V2 scDb and variants were analyzed against KRAS$^{WT}_{7-16}$-HLA-A*03:01 (Fig. 6b). The unique peptide binding groove residues in HLA-A*11:01 could alter the conformation of the KRAS$^{G12V}_{7-16}$ peptide and prevent its recognition by V2 scDb.

## Discussion

The KRAS$^{G12V}$ mutation is one of the most frequent mutations in solid organ tumors, and is presented by HLA-A*03:01, the 6th most common HLA class I allele in the United States[1]. We previously reported the discovery of a bispecific antibody, termed V2, which can specifically bind to the KRAS$^{G12V}$-HLA-A*03:01 neoantigen and drive T-cell-mediated killing of KRAS$^{G12V}$-containing cancer cells without non-specific activity against wild-type KRAS or other codon 12 mutations[19]. Here, we describe the structure of the V2 TCRm in complex with KRAS$^{G12V}$-HLA-A*03:01, and our work to characterize and improve interactions between the V2 bispecific and the KRAS$^{G12V}$ peptide.

This report includes the cryo-EM structure of an antibody fragment binding a MANA pHLA target, and the structures of the KRAS$^{WT/G12V}_{7-16}$ peptides presented by HLA-A*03:01 with or without an antibody in complex. Despite the inherent flexibility and dynamic structures of full-length IgG's, cryo-EM allows for high-resolution visualization of a Fab-HLA interaction without the limitation of crystallization artifacts that could influence protein conformation. While the KRAS$^{WT}$-HLA-A*03:01 peptide is involved in crystal packing, the molecular dynamics simulations conducted in this work support the observed KRAS$^{WT}$ peptide conformation and the conformational flexibility of the KRAS$^{G12V}$ peptide upon V2-Fab binding. Moreover, such techniques and approaches can be applied to systems where conformational flexibility may be

**Fig. 6 | Binding affinity and biophysical characterization of the V2 variant scDbs. a** scDbs were applied to ELISA plates coated with CD3ε/CD3δ heterodimer, KRAS$^{G12V}$-HLA-A*03:01, or the KRAS$^{G12WT}$-HLA-A*03:01 monomer. Relative binding was calculated as the ratio of KRAS$^{G12V}$–pHLA binding:CD3 binding for each scDb, and then reported as the percent of the original V2 scDb's relative binding. $n = 3$ for each target. **b** scDb binding to KRAS$^{G12V}$-HLA-A*3:01 (red) and KRAS$^{WT}$-HLA-A*3:01 (black) was evaluated by single-cycle kinetics SPR. There was negligible binding to KRAS$^{WT}$-pHLA (black line). The data for KRAS$^{G12V}$-HLA-A*3:01 was fit with two state binding kinetics (gray line) for **b** original V2 scDb ($K_D = 34$ nM), **c** F53W$^{L2}$ scDb ($K_D = 10.6$ nM), **d** V104N$^{H3}$ scDb ($K_D = 185.5$ nM) and **e** V104R$^{H3}$ scDb ($K_D = 6.6$ nM). V104R$^{H3}$ binds to the KRAS$^{WT}$-pHLA with a $K_D > 3.2$ μM. All sensorgrams are a representative experiment of $n = 3$ independent experiments.

important for selectivity, or in which crystallization artifacts may complicate structural interpretation.

Comparison of the KRAS$^{WT}$-HLA-A*03:01 structure with the V2-Fab/KRAS$^{G12V}$-HLA-A*03:01 complex structure suggests two possible co-operating mechanisms of selectivity. The first is the loss of a hydrophobic side chain in the wild-type form. The mutant valine side chain sticks out of the pHLA plane by ~3 Å, making hydrophobic interactions with Pro103$^{H3}$, Val104$^{H3}$, Tyr105$^{H3}$, and Phe53$^{L2}$ that are not possible for the glycine residue of the KRAS$^{WT}$ peptide. Valine at

position 12 provides a better and larger overall surface for contacts to be made with the V2-Fab CDRs. The second mechanism of selectivity suggested by the structures is a conformational change in the KRAS$^{G12V}$ peptide upon V2 binding to the pHLA complex. Unfortunately, we were unable to crystallize the KRAS$^{G12V}$-pHLA alone, but differential scanning fluorimetry, comparing the stability of the two monomers, suggests that the KRAS$^{WT}$-HLA-A*03:01 and KRAS$^{G12V}$-HLA-A*03:01 have similar structures. While the KRAS$^{WT}$ peptide has a greater buried surface area and more HLA interactions with peptide residues Val8-Gly13, the

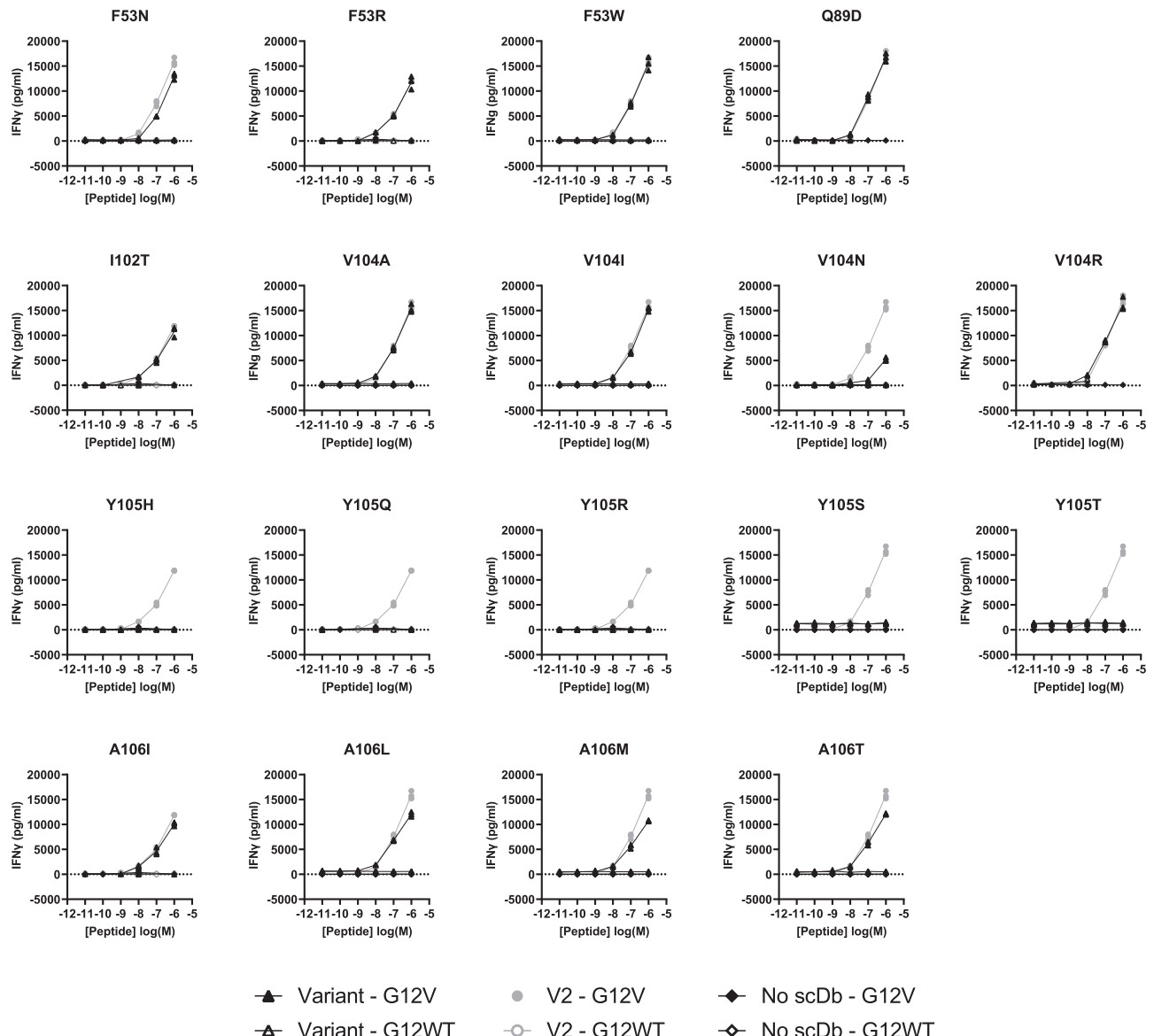

**Fig. 7 | Peptide pulsing co-culture assay.** V2 variants (triangles) were compared to the original V2 scDb (circle) in a peptide-pulsing co-culture with human T cells and T2A3 cells, with an E:T of 2:1 and 1 nM of scDb. T2A3 cells were pulsed with 10 μM to 10 nM of KRAS^G12V_7–16 10mer (solid) or KRAS^G12WT_7–16 10mer (open). After 16-20 h, cell culture supernatant was collected for detection of IFNγ by ELISA. Each variant was compared to V2 scDb in one experiment with *n* = 3 biologically independent samples for each co-culture condition; individual data points are plotted.

KRAS^G12V peptide in complex with V2 is less buried and has a unique hydrogen bond to the α2 in HLA-A\*03:01. We propose that V2 TCRm binding could drive a conformational change in the KRAS^G12V peptide that contributes to V2's specificity for the KRAS^G12V peptide over KRAS^WT. Others have shown that neoantigenic peptides can undergo conformational changes upon binding by TCRs. Sim et al. reported conformational changes in the peptide upon TCR binding to KRAS^G12D_10-19-HLA-C\*08:02, but not upon binding of a different TCR to KRAS^G12D_10-18-HLA-C\*08:02[23]. In addition, groups have reported examples where TCRs or TCRms reactive to mutant neoantigens exhibit strong specificity for mutant over wild-type peptides, despite minimal structural differences in the mutant vs. wild-type pHLA[35,43,44].

Various reports have shown that increasing bispecific antibody affinity generally leads to improved functional potency[20,28,36–40], with low-density targets requiring low nanomolar to picomolar affinity for optimal potency and specificity[37]. The design of improved pHLA targeting TCRs or TCRm antibodies can be complicated by the need to maintain specificity while improving affinity[45–47]. In general, affinity-

enhanced TCRs achieve this more efficiently through a broader, more balanced footprint of peptide and HLA interactions, while TCRm antibodies tend to concentrate binding to hotspots on the pHLA, leading to a greater degree of cross-reactivity[28]. The most specific TCRm antibodies make a larger proportion of interactions with the peptide compared with the HLA, and specifically with peptide side chains[28]. Modifications that increase binding affinity by simply adding interactions with the HLA can increase the potential for off-target binding to other pHLA. The V2 scDb is a particularly challenging example. It demonstrates hotspot binding at the C-terminus of the peptide with no interactions at the N-terminus. Furthermore, the majority of amino acids in the peptide are small, lack a side chain (glycine), or are hydrophobic[19]. This precludes introducing new hydrogen bonding interactions with the peptide side chains, instead limiting peptide interactions to the backbone or hydrophobic side-chain interactions. While small amino acids can be ideal for binding into small pockets created by the antibody, and driving selectivity by steric effects, many pHLA-TCR/TCRm interactions rely on polar

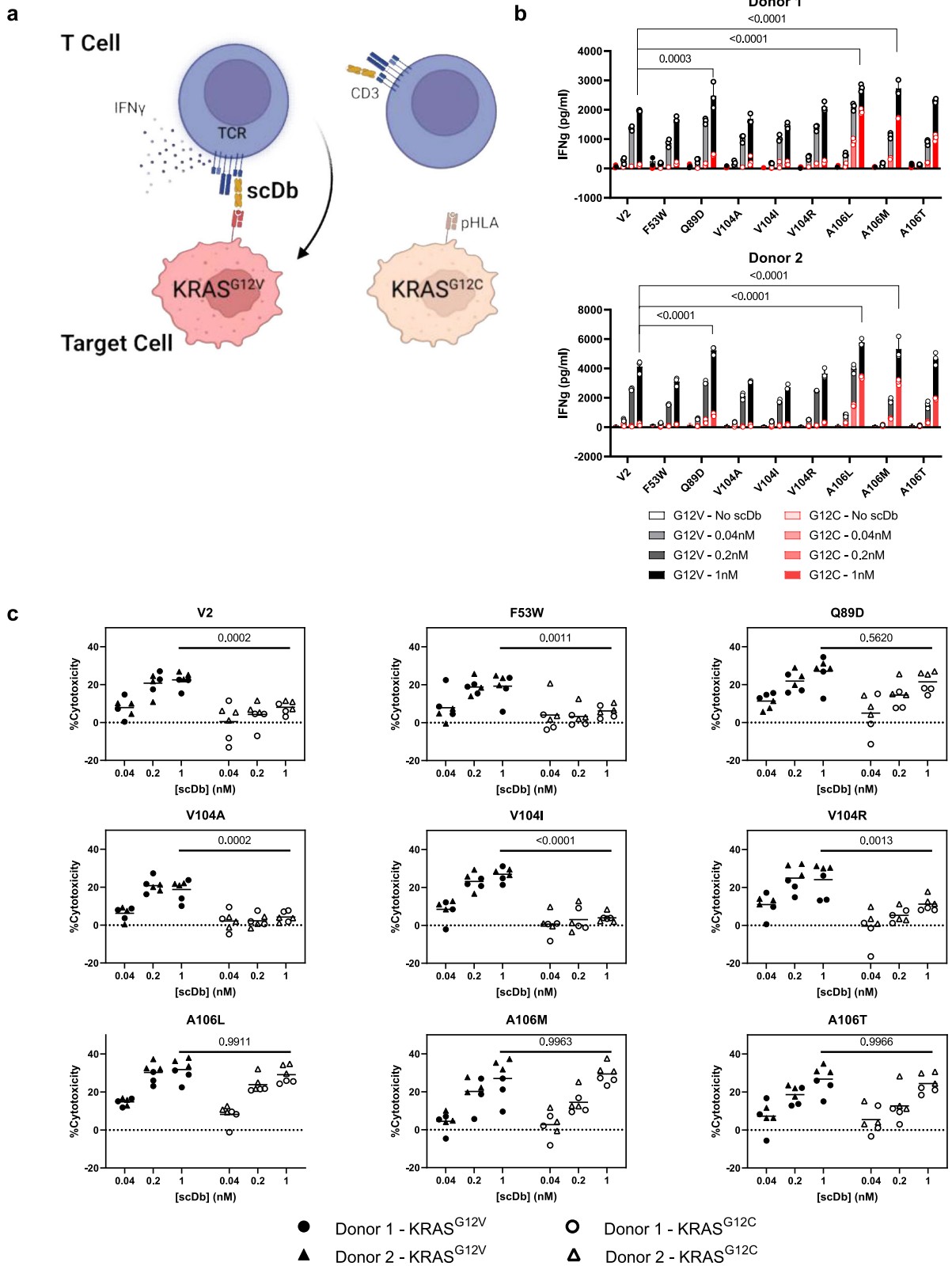

interactions[18,44,48]. Furthermore, only 3 of 6 CDRs are involved in the 26 peptide contacts, with most peptide contacts involving CDR[H3]. Overall, these features may limit V2 binding specificity by favoring interactions with the peptide main chain and HLA-A*03:01. The design space for improvements to V2 is further limited by the heavy tilt of the scDb, where approximately two-thirds of the total buried surface area is attributed to heavy chain interactions and HLA interactions.

Despite these limitations, we identified several V2 variants with higher relative binding by ELISA or measured affinity by SPR. These higher affinity variants, like F53W[L2] and V104R[H3], had maintained specificity against the KRAS[WT]-pHLA, but unfortunately failed to yield significantly improved sensitivity to low antigen levels, or substantially higher T cell activation, as measured by IFNγ release or cytotoxicity. One factor that may limit the benefit of higher affinity in this particular

**Fig. 8 | Reactivity of V2 variants to endogenous pHLA. a** Design of co-culture assay. T cells from two human donors were incubated with luciferase-expressing NCI-H358 clonal cell lines harboring the KRAS G12C or G12V mutation in the presence of varying concentrations of scDb for 20 h. Figure created with BioRender.com. **b** The concentration of IFNγ in the cell culture supernatant was measured by ELISA. IFNγ values (in pg/ml) for the G12V-containing condition (black) are overlaid with IFNγ values for the G12C-containing condition (red). Plotted data indicate mean ± SD, $n = 3$ biologically independent samples for each condition from one experiment. IFNγ values were compared to V2 scDb IFNγ at each scDb concentration in G12V conditions by two-way ANOVA with Dunnett's multiple comparisons test. A106M, A106L and Q89D yielded higher IFNγ than V2 at 1 nM scDb ($p < 0.0001$, <0.0001 and 0.0003, respectively). **c** Remaining viable cells were measured using the SteadyGlo luciferase assay, and cytotoxicity was calculated as (1 − (scDb condition/no scDb condition)) × 100%. Individual cytotoxicity values for each scDb are displayed for T cell donor 1 (circles) and donor 2 (triangles) by target cell KRAS mutation: KRAS$^{G12V}$ (black triangles and circles) or KRAS$^{G12C}$ (open triangles and circles). Grand mean cytotoxicity is indicated with a bar for each scDb concentration, $n = 3$ biologically independent samples per donor from one experiment. For each scDb, cytotoxicity at 1 nM was compared between the G12V- and G12C-containing conditions by two-way ANOVA with Šídák's multiple comparisons test pooling both donors ($n = 6$ per condition). V2, F53W, V104A, V104I, V104R had increased cytotoxicity against H358-KRAS$^{G12V}$ compared to H358-KRAS$^{G12C}$ ($p = 0.0002$, 0.0011, 0.0002, <0.0001, and 0.0013, respectively). Q89D, A106L, A106M and A106T did not have higher cytotoxicity against the KRAS$^{G12V}$ line ($p = 0.562$, 0.9911, 0.9963, 0.9966, respectively).

case is the strong tilt and C-terminal shift of the V2 scDb on the pHLA. This binding orientation could impact how bispecific antibodies crosslink and activate T cells upon target recognition, limiting the benefit of improved binding affinity.

Targeting the same peptide across multiple HLA types offers an appealing way to expand the population of patients that could benefit from a KRAS$^{G12V}_{7–16}$-targeting therapy. But in our case, SPR comparison of scDb binding to KRAS$^{G12V}$-HLA-A*03:01 and KRAS$^{G12V}$-HLA-A*11:01 indicates that all of the variants are specific for the KRAS$^{G12V}$ peptide presented on HLA-A*03:01, and do not have substantial binding to the HLA-A*11:01 complexes. Identifying an antibody that is highly specific for the KRAS$^{G12V}$ peptide across multiple HLAs would require binding reliant on specific peptide-antibody interactions, presumably with less contribution from HLA-antibody interactions. Incorporation of a positive selection step for KRAS$^{G12V}$-HLA-A*11:01 in the original or affinity maturation panning stages may allow identification of such a clone.

Our characterization of the antibodies and pHLA complexes described above offers insights into how highly hydrophobic peptide neoantigens can be targeted with T cell-redirecting therapies. Our data illustrate the potential for antibodies to recognize hydrophobic peptides presented on HLA molecules with high affinity and specificity.

## Methods

### Plasmid availability
Plasmids are available under a material transfer agreement with Johns Hopkins university by request.

### Expression, purification and refolding of KRAS$^{G12V/WT}$-HLA-A*03:01
Plasmids for HLA-A*03:01 and β2M were received from the NIH Tetramer Facility (Atlanta, GA) and separately transformed into BL21(DE3) cells. Both were subsequently expressed in inclusion bodies using auto-induction media as previously described[49–51]. Briefly, a small 2 ml culture was grown in ZYP-0.8 G medium containing N-Z-amine tryptone, yeast extract, 1 mM MgSO₄, 0.8% glucose and 1× of 20× NPS solution (0.5 M (NH₄)₂SO₄, 1 M KH₂PO₄, 1 M Na₂HPO₄) for 6–8 h at 37 °C. Once the solution was turbid, 200 μL of the culture was diluted into 400 mL of auto-induction media, ZY-5052, consisting of N-Z-amine tryptone, yeast extract, 1 mM MgSO₄, 1× of 20× NPS solution and 1× of 50 × 5052 solution (0.5 % glycerol, 0.05% glucose, 0.2% α-lactose) and incubated overnight at 37 °C. Purification of the HLA-A*03:01 and β2M inclusion bodies was achieved with a series of detergent washes followed by solubilization with 8 M urea. Refolding of the HLA-A*03:01, β2M, and KRAS$^{G12}$ peptide (either WT or mutant) was performed as previously described[49–51]. Briefly, urea solubilized HLA-A*03:01 (2000 nmol) and β2M (4000 nmol) were combined in 1 L refolding buffer (100 mM Tris pH 8.3, 400 mM L-arginine, 2 mM EDTA, 5 mM reduced glutathione, 0.5 mM oxidized glutathione, 2 mM PMSF) with 30 mg of either the mutant KRAS$^{G12V}$ peptide (aa 7–16, VVVGA**V**GVGK) or wild-type KRAS$^{WT}$ peptide (aa 7–16, VVVGA**G**GVGK) dissolved in DMSO. The resultant solution was stirred at 4 °C for 2 days, with two further 2000 nmol additions of HLA-A*03:01 on day 2, concentrated to 10 ml, and purified by size exclusion chromatography on a HiLoad 26/60 Superdex 75 Prep grade column (Cytiva, 28989334). Purified KRAS$^{WT}$/HLA-A*03:01 was concentrated to 12 mg/ml and stored at −80 °C until use. For incubation with the V2-IgG, purified KRAS$^{G12V}$/HLA-A*03:01 was concentrated to ~1–2 mg/ml and stored at −80 °C until use.

### Production and purification of the V2-IgG full-length antibody
The $V_H$ and $V_L$ sequences of the V2 scFv were grafted onto the respective constant chains of the Herceptin/trastuzumab antibody. Both the light and heavy chains were separately cloned into a pcDNA3.4 vector with a leader peptide mouse IgKVIII sequence (Thermo Fisher Scientific, Waltham, MA). For a 1 L expression, the light and heavy chain plasmids were co-transfected at a 1:1 ratio into Expi293 cells (Thermo Fisher Scientific, A14527) at a concentration of $4.6 × 10^6$ cells per ml. The supernatant was harvested 6 days post-transfection, and the full-length V2-IgG was purified via protein A chromatography using a HiTrap protein A HP column. The V2-IgG was eluted using 100 mM citric acid, pH 3. The protein A fractions containing pure V2-IgG were pooled, evaluated by Coomassie Blue staining, dialyzed into PBS, and stored at −80 °C (GeneArt, Thermo Fisher Scientific, Regensburg, Germany).

### Preparation and purification of the V2-IgG/KRAS$^{G12V}$-HLA-A*03:01 complex
The V2-IgG was mixed at a 1:3 molar ratio with KRAS$^{G12V}$-HLA-A*03:01 and incubated at 4 °C overnight. The V2-IgG/KRAS$^{G12V}$-HLA-A*03:01 mixture was evaluated by size exclusion chromatography on a Superose™ 6 10/30 GL (Cytiva, 17517201) with the following buffer: 50 mM TRIS, pH 8, 200 mM NaCl. The fractions of ~ 98% pure V2-IgG/pHLA-A*03:01 complex were pooled and concentrated to ~0.5–2 mg/ml.

### Crystallization, data collection and structure determination of KRAS$^{WT}$-HLA-A*03:01
Crystals of the KRAS$^{WT}$-HLA-A*03:01 were grown by vapor diffusion in hanging drops set up with a TTP mosquito robot with a reservoir solution of 0.2 M ammonium sulfate, 0.1 M sodium cacodylate pH 6.5, 30% w/v PEG 8000. Crystals were flash-cooled in mother liquor. Data were collected at National Synchrotron Light Source-II at beamline 17-ID-2 (FMX) on a Dectris EIGER X 16 M detector using LSDCGui and a vector of 90 μm over 360°[52]. Only frames for the first 180° were used for the final dataset. The dataset was indexed, integrated and scaled using XDS[53]. Thin needle crystals of KRAS$^{WT}$-HLA-A*03:01 diffracted to 2.6 Å. The structure was determined by molecular replacement with MolRep[54] using PDB ID 6O9B as the search model[55]. The data were refined to a final resolution of 2.6 Å using iterative rounds of refinement with REFMAC5[56,57], Phenix[58] and manual rebuilding in Coot[59]. Structures were validated using Coot and PDB Deposition tools. The model has 97.9% of the residues in preferred and 2.1% in allowed

regions according to Ramachandran statistics (Table 2). Figures were rendered in PyMOL (v2.4.2, Schrödinger, LLC, New York, NY).

## Cryo-electron microscopy specimen preparation and data collection of V2-IgG/KRAS$^{G12V}$-HLA-A*03:01

For cryo-EM, a 3 µl solution of mixed V2-IgG/KRAS$^{G12V}$-HLA-A*03:01 (concentration 0.15-0.2 mg/ml) and detergent FC14 (final concentration 0.05 mM) was applied to glow-discharged (60 s, 15 mA, easyGlow) Quantifoil UltraAu grids (300 mesh, R1.2/1.3). The grids were blotted for 1.5 s at 100% humidity and plunge-frozen into liquid ethane using a Thermo Fisher Vitrobot Mark IV. Cryo-EM data were collected at the Cryo-EM facility at the HHMI Janelia Research Campus. Images were collected on a 300 kV Titan Krios cryo-EM equipped with a Gatan Bioquantum energy filter and a Gatan K3 detector. Images were taken on the K3 camera in dose-fractionation mode at a nominal magnification of ×105,000, corresponding to 0.844 Å per physical pixel (0.422 Å per super-resolution pixel). The dose rate on the specimen was set to be 14 electrons/Å$^2$/s and the total exposure time was 4.27 s, resulting in a total dose of 60 electrons/Å$^2$. With dose-fractionation set at 0.0855 s/frame, each movie series contained 50 frames and each frame received a dose of 1.2 electrons/ Å$^2$. Fully automated data collection was carried out using SerialEM software[60] with a nominal defocus range set from −0.8 to −2.0 µm.

## Cryo-EM image processing

Movie frames were aligned and corrected for beam-induced motion using the Relion implementation of MotionCor2[61] with 5 by 5 patches. Contrast transfer functions (CTF) were estimated using CTFFIND-4.1[62]. Micrographs with a CTF fit of better than 5.0 Å were used for further processing. Initial rounds of blob-based picking were used to generate templates via 2D classification in cryoSPARC[63]. Templates were imported in Relion 3.1[64] and used for template picking. Using a lenient threshold, 3,603,116 particles were picked and extracted from the micrographs (640 box size binned to 80). Particles were transferred to cryoSPARC and multiple rounds of 2D classification were used to clean up the particle set to 367,584 particles. These particles were used to generate an ab initio model in cryoSPARC, which was then refined with Homogeneous Refinement, and the particles and model were imported back into Relion. The particles were re-extracted with a box-size of 640 binned to 320 (pixel size of 0.844 Å/pixel). 3D classification into 4 classes yielded a single high-resolution class with 168,185 particles, which were refined to 4.03 Å. Two rounds of per-particle CTF refinement and particle polishing in Relion were followed by 3D refinement to 3.37 Å. Focused 3D classification without alignment into 4 classes, using a mask encompassing the variable region of the antibody and the peptide binding cleft of the HLA, yielded a single good class with 116,685 particles. A value of T = 4 was used for both 3D classification steps. 3D refinement with SIDESPLITTER[65] in Relion produced a 3.14 Å map, as assessed by an FSC threshold of 0.143. Local resolution was calculated with MonoRes[66], and local map sharpening was done with Local Deblur[67].

## Model building and structure refinement for V2-Fab/ KRAS$^{G12V}$-HLA-A*03:01

The coordinates of the KRAS$^{WT}$-HLA-A*03:01 monomer (determined in this paper, PDB ID 8DVG and PDB entry 7KGU (2Q1-Fab) were roughly fitted into the EM map using ChimeraX[68]. Manual model building was carried out in Coot[59], where the sequence of each component was in silico mutated to the corresponding residues of V2-Fab and KRAS$^{G12V}$-HLA-A*03:01. After real-space refinement with global minimization and rigid body constraints in Phenix[69], the model was further rebuilt and refined iteratively using Coot, and was validated using PDB deposition tools. Refinement statistics are summarized in Table 1. The structure and cryo-EM maps were visualized using UCSF Chimera X and Pymol (Schrodinger, LLC). Secondary structure elements were assigned using

DSSP in Phenix. Buried areas were calculated with PDBePISA[70]. The docking angle that determines the relative orientation between the pHLA and the V2-Fab was calculated by the web server TCR3d[71,72].

Molecular interactions were calculated using CONTACTS in the CCP4 suite[56] and evaluated with LigPlot + v2.2[73].

## Differential scanning fluorimetry

Thermal stability of the KRAS$^{G12V}$-HLA-A*03:01 and KRAS$^{WT}$-HLA-A*03:01 monomers were evaluated by a differential scanning fluorimetry (DSF) assay, which monitors the fluorescence of a dye that binds to the hydrophobic region of a protein as it becomes exposed upon temperature-induced denaturation[74,75]. Reaction mixture (20 µL) was set up in a white low-profile 96-well, unskirted polymerase chain reaction plate (BioRad, MLL9651) by mixing 2 µL of purified pHLAs at a concentration of 1 mg/mL (final concentration 0.1 mg/mL) with 2 µL of 50X SYPRO orange dye (Invitrogen, S6650, 5× final concentration) in 50 mM TRIS, pH 8, 200 mM NaCl. The plate was sealed with an optically transparent film (Thermo Fisher Scientific, 4311971) and centrifuged for 1000 × g for 30 s. Thermal scanning was performed from 25 to 100 °C (1 °C/min temperature gradient) using a CFX9 Connect real-time polymerase chain reaction instrument (BioRad). Protein unfolding/melting temperature T$_m$ was calculated from the minimum value of the negative first derivative of the melt curve using CFX Manager software (BioRad).

For DSF of the original V2 scDb and variants, the scDbs were dialyzed into PBS overnight, and their resulting concentrations ranged from 2.42 mg/ml to 3.69 mg/ml. 2 µL of each scDb were put in individual wells of a clear semi-skirted 96-well PCR plate and diluted to 20 µL with 16 µL of water and 2 µL of dye. Samples were performed in triplicate, and the plate was sealed with Bio-Rad Sealing Tape for Optical Assays and then centrifuged at 500 × g for 1 min. Thermal scanning was performed from 25 to 100 °C (1 °C/min temperature gradient) using a CFX9 Connect real-time polymerase chain reaction instrument (BioRad). Protein unfolding/melting temperature T$_m$ was calculated from the minimum value of the negative first derivative of the melt curve using CFX Manager software (BioRad). All resulting graphs were created with GraphPad Prism 9.

## Surface plasmon resonance affinity measurements

Biotinylated pHLAs (KRAS$^{G12V}$-HLA-A*03:01, KRAS$^{WT}$-HLA-A*03:01, KRAS$^{G12V}$-HLA-A*011:01, KRAS$^{WT}$-HLA-A*011:01), and V2 scDb binding experiments were performed at 25 °C using a Biacore T200 SPR instrument (Cytiva) in HBS-P buffer. Approximately 160 response units (RU) of biotinylated pHLAs were captured in different flow cells (Fc) using a streptavidin chip. Single-cycle kinetics were performed by injecting increasing concentrations (100, 200, 400, 800, 3200 nM) of purified V2-scDb, which was flowed over all flow cells. Binding responses for kinetic analysis were both blank- and reference-subtracted[76]. Both binding curves were fit with a two-state model using Biacore Insight evaluation software.

## Cell lines and primary cells

All cells were cultured at 37 °C with 5% CO$_2$. T2A3 (from Eric Lutz and Elizabeth Jaffee, JHU), Jurkat (ATCC TIB-152, Manassas, VA), Raji (ATCC CCL-86), and NCI-H358 (ATCC CRL-5807) cells were cultured in RPMI-1640 (ATCC, Manassas, VA) with 10% HyClone FBS (GE Healthcare, Chicago, IL) and 1% Penicillin-Streptomycin (Gibco, Thermo Fisher Scientific). Hs766T (ATCC HTB-134) were cultured in DMEM (Gibco, Thermo Fisher Scientific) supplemented with 10% HyClone FBS and 1% Penicillin-Streptomycin. RPMI-6666 (ATCC CCL-113) cells were cultured in RPMI-1640 supplemented with 20% HyClone FBS and 1% penicillin−streptomycin. HEK293FT (Thermo Fisher Scientific R70007) cells were cultured in DMEM containing 4.5 g/L glucose, glutaMAX supplement (4 mM L-alanyl-glutamine), and 110 mg/L sodium pyruvate (Gibco, Thermo Fisher Scientific #10569010) with 10% HyClone FBS, 1%

 

penicillin–streptomycin, 0.1 mM MEM Non-Essential Amino Acids (Gibco, Thermo Fisher Scientific), 2 mM GlutaMAX (Gibco, Thermo Fisher Scientific), and 500 μg/ml Geneticin (Gibco, Thermo Fisher Scientific), with a final glutamine concentration of 6 mM. The NCI-H358 isogenic cell lines were previously derived[19]. Briefly, parental NCI-H358 cells and KRAS G12V CRISPR knock-in pools were single cell cloned, selected for similar growth characteristics and transduced with a luciferase-GFP-containing virus (Perkin Elmer CLS960003). Luciferase-GFP-positive cells were isolated by flow cytometric sorting.

Human peripheral blood cells were obtained from fresh leukopaks (Stem Cell Technologies, Cambridge, MA) and purified via density gradient centrifugation with Ficoll Paque Plus (GE Healthcare). Red blood cells were removed using ACK lysing buffer (Quality Biological, Gaithersburg, MD). Purified PBMCs were frozen in HyClone FBS containing 10% DMSO (Sigma Aldrich, St. Louis, MO) and stored at −150 °C. T cells were expanded from frozen PBMCs with 15 ng/ml anti-CD3 antibody (clone OKT3, 317347, Biolegend, San Diego, CA) in T cell culture medium: RPMI-1640 (ATCC, Manassas, VA) containing 10% HyClone FBS (GE Healthcare), 1% penicillin–streptomycin (Thermo Fisher Scientific), 100 IU/ml recombinant human IL-2 (aldesleukin, Prometheus Therapeutics and Diagnostics, San Diego, CA), and 5 ng/ml recombinant human IL-7 (Biolegend,San Diego, CA). T cell culture medium was changed every 3 days.

### Peptides and HLA monomers
Peptides were purchased from Peptide 2.0 (Chantilly, VA) at ≥ 90% purity: KRAS$^{G12V}$[7–16] VVVGAVGVGK, KRAS$^{WT}$[7–16] VVVGAGGVGK. Biotinylated monomers (bKRAS$^{G12V}$-HLA-A*03:01, bKRAS$^{WT}$-HLA-A*03:01, bKRAS$^{G12V}$-HLA-A*11:01, and bKRAS$^{WT}$-HLA-A*11:01) were produced by the Fred Hutchinson Immune Monitoring Core using supplied peptides from Peptide 2.0 (Fred Hutchinson Cancer Center, Seattle, WA).

### Affinity maturation library design and construction
Complementarity determining regions (CDR) in the V2 scDb were identified by protein sequence annotation using the AbYsis tool[77]. In some cases, CDR regions were trimmed or expanded to include unusual residues (low frequency in *Homo sapiens* sequences in the AbYsis database) or exclude residues with low diversity (2–3 total amino acids represented in >90% of *Homo sapiens* sequences in the AbYsis database). Sixty-one sites across the 6 CDRs, with 19 non-wild type amino acids per site, were selected for the affinity maturation library, for a total of 1159 variants. The variant library was directly synthesized and cloned into the pADL-10b phagemid vector (Antibody Design Labs, San Diego, CA) by Twist Bioscience (San Francisco, CA). Library diversity was confirmed by next generation sequencing performed by Twist Bioscience.

### Generating the affinity maturation phage library
The 1159 variants in the V2 affinity maturation library were pooled and reconstituted in sterile water. 30 μl (60 ng) of library DNA was combined with 20 μl SS320 cells on ice, and divided into two 25 μl aliquots for electroporation with the Gene Pulser electroporation system (Bio-Rad, Hercules, CA). Cells were electroporated (200 ohms, 25 microFarads, 1.8 kV) and immediately supplemented with a total of 660 μl recovery media (Lucigen) and incubated for 45 min at 37 °C with shaking at 250 rpm. After recovery, 7 μl of cells were used to determine transformation efficiency by serial dilution titering. The remaining cell volume was plated on 2xYT medium (Sigma-Aldrich) containing 100 μg/ml carbenicillin and 20% D-glucose (Sigma) in a 24 × 24 cm plate. Plated cells were incubated for 6 h at 37 °C and then stored overnight at 4 °C. The plated cells were scraped into 2XYT media containing 100 μg/ml carbenicillin. The cell suspension was brought to approximately 5 × 10$^6$ transformants per ml in 2XYT + 100 μg/ml

carbenicillin containing 20% glycerol, and snap frozen in 1 ml aliquots for storage at −80 °C.

A phage reference library was produced from 1 ml glycerol stock (approximately 5 × 10$^6$ transformants) by incubating stored cells in 50 ml 2xYT medium containing 100 μg/ml carbenicillin and 2% D-glucose at 37 °C, with 225 rpm shaking until the culture reached an optical density at 600 nm (OD$_{600}$) of 0.3-0.5. When cells reached an OD$_{600}$ of 0.3–0.5, 100 μl of M13K07 helper phage (2 × 10$^{12}$ pfu/ml, Antibody Design Labs) was added to the culture and incubated for 1 h at 37 °C with 225 rpm shaking. Cells were centrifuged at 4000 × $g$ for 8 min and resuspended in 2xYT media containing 100 μg/ml carbenicillin and 50 μg/ml kanamycin. Cells were cultured overnight at 30 °C with 250 rpm shaking. The culture was centrifuged at 5000 × $g$ for 10 min. The supernatant was collected and cleared of residual bacteria by centrifugation at 12,000 × $g$ for 10 min. The resulting phage-laden supernatant was added to 20% polyethylene glycol 8000 (PEG, Sigma-Aldrich) in 2.5 M NaCl at a 4:1 ratio (4% PEG final) and incubated on ice for 30-60 min to precipitate phage. Phage was collected by centrifugation at 12,000 × $g$ at 4 °C for 40 min, and the resulting phage pellet was resuspended in 1× Tris-buffered saline (TBS, 25 mM Tris-HCl, 150 mM NaCl, pH 7.5) containing 2 mM EDTA (TBSE) and 0.1% sodium azide. This reference library phage was stored in 50% glycerol at −20 °C. Reference library phage titer was determined by infecting SS320 cells at an OD$_{600}$ of 0.3–5 with serial dilutions of phage and growing overnight on 2XYT medium agarose plates with 100 μg/ml carbenicillin and 2% glucose. The V2 phage reference library had a titer of 5.18 ×10$^{10}$ transformants/ml.

### Selection of phage clones by panning
Reference library phage was used to produce the panning phage library. SS320 competent cells were cultured in 2XYT media containing 2% glucose and 20 μg/ml tetracycline to reach an OD$_{600}$ of 0.3–0.5. 450 μl reference phage library and 50 μl M13K07 helper phage were added to the SS320 culture and incubated for 1 h at 37 °C. Cells were collected by centrifugation at 3000 × $g$ for 10 min and resuspended in 100 ml 2XYT medium containing 100 μg/ml carbenicillin, 50 μg/ml kanamycin, and 10 μM isopropyl β-D-1-thiogalactopyranoside (IPTG). Cells were incubated overnight at 37 °C with 250 rpm shaking to generate the panning phage library. The panning phage library was precipitated in the same method as the reference phage library, except the final precipitated phage pellet was resuspended in 2 ml TBSE containing 0.1% sodium azide. The V2 panning library titer was 8.45 × 10$^{10}$ transformants/ml.

Panning for V2 variants specific for the KRAS$^{G12V}$[7–16] 10mer peptide on HLA-A*03:01 were selected similarly to previously described methods[19]. In the enrichment phase, non-specific phage were removed from a starting phage pool of 1.78 × 10$^{11}$ phage by negative selection against streptavidin-coated M280 Dynabeads (Thermo Fisher) and 1 mg/ml streptavidin in an overnight incubation at 4 °C on a rotator. Supernatant from negative selection was transferred to a tube containing 30 μl M280 beads pre-coated with the KRAS$^{G12V}$/HLA-A3 monomer (1 μg monomer per tube) and 20 μg streptavidin for positive selection. After 1 h of positive selection on a rotator at room temperature, phage-laden pMHC-coated M280 beads were washed ten times using 1 ml TBST (1× TBS containing 0.5% Tween20 (Sigma-Aldrich)). The remaining bound phage was eluted by incubated beads in 1 ml 0.2 M glycine pH 2.2 for up to 10 min on a rotator. The elution mixture was transferred to a new tube containing 150 μl Tris pH 9 to neutralize the solution. 10 μl M13K07 helper phage and 1 ml eluted phage were used to infect 9 ml SS320 cells (OD600 0.3–0.5) in 2XYT medium containing 100 μg/ml carbenicilin and 2% glucose for 1 h at 37 °C with shaking at 250 rpm. Cells were collected by centrifugation and resuspended in 2XYT medium containing 100 μg/ml carbenicillin, 50 μg/ml kanamycin and 10 μM IPTG. Phage was grown overnight and

precipitated as described above. This round 1 phage pellet was resuspended in 1 ml TBS containing 2 mM EDTA and 0.1% sodium azide.

For rounds 2-5 of panning, decreasing starting amounts of phage were used for each round of negative selection: 10%, 1%, 0.1%, 0.02% of the total phage eluted in the previous round, respectively. Negative selection rounds 2-5 included incubation with $\geq 5 \times 10^7$ pooled HLA-A3+, KRAS$^{G12V}$− cells (RPMI-6666, T2A3, Jurkat, Raji, H358, Hs766T, and 293FT) at 4 °C for 16–24 h, and incubation with M280 beads coated with 1 µg monomers: KRAS$^{WT}$[7–16]/HLA-A3 or unrelated EGFR-derived peptide/HLA-A3 monomers (AIKTSPKANK, KELREATSPK), for an additional 16–24 h. After negative selection, unbound phage supernatant was transferred to new tube containing KRAS$^{G12V}$[7–16]/A3 monomer bound to M280 beads, and incubated for 1 h at room temperature. All selection incubations used a tube rotator. For positive selection in rounds 2–5, phage was selected using 0.5 µg monomer for rounds 2–4, and 0.25 µg for round 5. After positive selection, beads were washed 4 times with 1 ml TBST, and then incubated in a fifth wash of 1 ml overnight at room for rounds 2–5. An additional 5 washes (for 10 total) were performed using 1 ml TBST before proceeding to elute, amplify and precipitate the phage as described for round 1. All phage was resuspended in 1 ml TBS containing 2 mM EDTA and 0.1% sodium azide. After 5 rounds of negative and positive selection, the titer for phage from rounds 4 and 5 was determined by serially diluting the phage and infecting SS320 cells (OD600 0.3–0.5) with diluted phage. SS320 cells were incubated overnight on agarose 2xYT plates containing 100 µg/ml carbenicillin and 2% D-glucose. The titers for V2 Round 4 and V2 Round 5 were $5.3 \times 10^{11}$ and $3.5 \times 10^{11}$ transformants/ml, respectively.

### Selecting clones for scFv characterization
Single colonies from the Round 4 and Round 5 titering plates were inoculated into 200 µl 2XYT medium containing 100 µg/ml carbenicillin and 2% glucose in a 2-ml deepwell plate (Thermo Fisher Scientific). Bacteria was cultured for 3 h at 37 °C with shaking at 250 rpm. After 3 h, cells were infected with M13K07 helper phage ($4 \times 10^8$ per well) and incubated for 1 h at 37 °C with shaking at 250 rpm. Culture media was removed after centrifugation at $3000 \times g$ for 10 min and replaced with 300 µl fresh 2XYT medium containing 100 µg/ml carbenicillin, 50 µg/ml kanamycin, and 20 µM IPTG. Cells were cultured overnight at 30 °C with shaking at 250 rpm and then pelleted by centrifugation at $3000 \times g$ for 10 min. Plates with phage-laden supernatant were stored at 4 °C for downstream analysis.

### PCR and Sanger sequencing
To determine the variants present in the colonies picked from rounds 4 and 5 of panning, we PCR amplified the scFv region from 1 µl of each monoclonal phage well using Q5 Hot Start Hi Fidelity 2X Master Mix (New England BioLabs) and the following primers: Amplification Forward GGCCATGGCAGATATTCAGA, Amplification Reverse CCGGGC CTTTATCATCATC. Amplicons were sequenced with the following primer by Genewiz (South Plainfield, NJ): GGCCATGGCAGATAT TCAGA. Sequences were trimmed to flank the CDRs using the DNA Baser Sequence Assemble v4 software (Arges, Romania) and clustered (100% sequence identity) using the CD-HIT Suite[78,79]. Unique phage clones were used for downstream characterization.

### Monoclonal phage ELISAs
Streptavidin-coated plates (R&D Systems, Minneapolis, MN) were coated with 50 µl of 1 µg/ml biotinylated KRAS$^{G12V}$[7–16]/HLA-A3 monomer or the KRAS$^{WT}$[7–16]/HLA-A3 monomer in BAE buffer (PBS containing 0.5% bovine serum albumin (Sigma), 2 mM EDTA (Thermo) and 0.1% sodium azide) for 1 h at room temperature. Plates were washed 6 times with 1X TBST (J77500-K8, Thermo Fisher Scientific) using a 405 TS microplate washer (BioTek, Winooski, VT). Phage-laden supernatant from the single colony cultures was diluted 5× with 1× TBST and added to the monomer-coated plates for a 2 h incubation at

room temperature. After 2 h, the plates were washed 6 times with 1× TBST. Plates were then incubated for 1 h at room temperature with a polyclonal rabbit anti-fd/M13 bacteriophage antibody (NB100-1633, Novus, Centennial, CO) diluted 1:5000 in 1× TBST. Plates were washed 6 times with TBST. Plates were then incubated with goat anti-rabbit IgG (H + L) Secondary (HRP) antibody (NB7160, Novus, Centennial, CO) diluted 1:10,000 in 1X TBST for 1 h at room temperature. Plates were washed 6 times with 1X TBST. Bound phage was detected by adding 3, 3′, 5, 5′-tetramethylbenzidine (TMB) substrate (BioLegend) and quenching the color formation reaction with 1 N sulfuric acid (Fisher Scientific). Absorbance was measured at 450 and 540 nm using a Synergy H1 Multi-Mode Reader (BioTek). Reported A$_{450}$ values include an A$_{540}$ correction (subtract A$_{540}$ from A$_{450}$ for each well). All samples were loaded in duplicate.

### scDb production
Plasmids encoding the single amino acid variants of the V2 single chain diabody were generated by site-directed mutagenesis using NEBaseChanger (New England Biolabs, Ispwich, MA) for primer design, and the Q5 Site Directed Mutagenesis Kit (New England Biolabs) following the manufacturer's protocol. All single chain diabodies were expressed using the pcDNA3.4 vector (Thermo Fisher Scientific) with an N-terminal IL-2 signal sequence and C-terminal 6X His tag, as previously reported[19]. For monomer ELISAs and co-culture testing, V2 scDb and its variants were expressed in HEK293FT cells. 20 µg of plasmid DNA was transfected into a T75 flask (Corning, Corning, NY) of HEK293FT cells using the Lipofectamine 3000 reagent kit (Thermo Fisher Scientific) following the manufacturer's protocol. After 5 days, cell culture supernatant was collected, cleared by centrifugation, and incubated with 25-50 µl of HisPur Ni-NTA Resin (Thermo Fisher Scientific) overnight on a rotator at 4 °C. Resin beads were collected by centrifugation and transferred to a Pierce Micro-Spin Column (Thermo Fisher Scientific). Residual cell culture supernatant was removed by centrifugation at $1000 \times g$ for 1 min. Resin was washed 4 times using 1 resin-volume of 20 mM imidazole (GE Healthcare) in PBS (Gibco, Thermo Fisher), with centrifugation at $1000 \times g$ for 1 min. scDb was eluted using 1 resin-volume of 250 mM imidazole in PBS and desalted into 20 mM Tris-HCl 150 mM NaCl pH 9 using 0.5 ml 7 K MWCO Zeba Spin Desalting Columns (Thermo Fisher Scientific) following the manufacturer's protocol. scDb concentration was determined by gel quantification using 4-15% Mini-PROTEAN TGX Stain-Free gels (BioRad). For SPR, scDbs were expressed in Expi293 cells (Thermo Fisher Scientific, A14527) by GeneArt (Thermo Fisher Scientific) and purified with a HisTrap column (GE Healthcare) followed by size exclusion chromatography using a HiLoad Superdex 200 16/600 column (GE Healthcare). Purified scDbs were analyzed using a TSKgel G3000SWxl column (TOSOH Bioscience) with 50 mM sodium phosphate 300 mM sodium chloride running buffer, pH 7, and a flow rate of 1 ml/min. scDbs were quantified by spectrophotometry (Nanodrop, Thermo Fisher Scientific).

### scDb monomer ELISAs
The streptavidin-coated plates (R&D Systems, Minneapolis, MN) were coated with 50 µl of 0.5 µg/ml of biotinylated KRAS$^{G12V}$[7–16]/HLA-A3 monomer, 0.5 µg/ml biotinylated KRAS$^{WT}$[7–16]/HLA-A3 monomer, or 0.4 µg/ml biotinylated recombinant human CD3ε/CD3δ heterodimer (CDD-H82W6, Acro Biosystems, Newark, DE) in BAE buffer (PBS containing 0.5% bovine serum albumin [Sigma], 2 mM EDTA [Thermo] and 0.1% sodium azide) for 1 h at room temperature. Plates were washed 6 times with 1× TBST (J77500-K8, Thermo Fisher Scientific) using a 405 TS microplate washer (BioTek, Winooski, VT). scDb samples were diluted to 200 ng/ml in PBS and incubated in the coated plates for 1 h at room temperature. Plates were washed 6 times with TBST. 50 µl of 0.5 µg/ml recombinant Protein L (Pierce) was added to the well and incubated for 1 h at room temperature. Plates were

washed 6 times with TBST. Plates were incubated with 50 µl of 0.2 µg/ml chicken anti-Protein L HRP antibody (ab63506, Abcam) for 1 h at room temperature and then washed 6 times with TBST. 50 µl of TMB substrate (Biolegend) was added to each well and quenched with 50 µl 1 N sulfuric acid (Fisher Scientific). Absorbance was measured at 450 and 540 nm using a Synergy H1 Multi-Mode Reader (BioTek). Reported $A_{450}$ values include an $A_{540}$ correction (subtract $A_{540}$ from $A_{450}$ for each well). All samples were measured in triplicate.

### Peptide pulsing co-culture

T2A3 cells were pulsed at $10^6$ cells/ml with $KRAS^{G12V}$[7–16] or $KRAS^{WT}$[7–16] 10mers in serum-free RPMI-1640 (ATCC) for 1 h at 37 °C and 5% $CO_2$. Pulsed cells were washed with complete media (RPMI-1640 supplemented with 10% FBS and 1% penicillin-streptomycin) before addition to a 96-well tissue culture-treated flat-bottom plate. 15,000 to 25,000 pulsed T2A3 cells were co-cultured with 1 nM scDb (50 ng/ml) and 30,000 to 50,000 T cells for an effector to target ratio (E:T) of 2:1 in a total volume of 100 µl. T cells from a single donor were used at least two weeks after expansion from PBMCs with OKT3. The co-culture was incubated for 18-21 h, and cell culture supernatants were collected and frozen for downstream analysis. Frozen supernatants were thawed for IFNγ measurement using the Human IFNγ Quantikine ELISA kit (DIF50C R&D). Absorbance was measured at 450 and 540 nm using a Synergy H1 Multi-Mode Reader (BioTek). IFNγ concentration was determined using a 4-Parameter Log fit and $A_{450s}$ value with an $A_{540}$ correction (subtract $A_{540}$ from $A_{450}$ for each well). All experiments were performed in triplicate.

### Isogenic cell co-culture

scDbs and T cells were co-cultured with an isogenic pair of luciferase-expressing clonal H358 cell lines as previously described: a parental clone (KRAS G12C/HLA-A3) and a CRISPR-edited KRAS G12V knock-in clone[19]. T cells from two donors were used 6 weeks after expansion from PBMCs with OKT3. In a white opaque flat-bottom tissue-culture treated plate (Thermo Fisher Scientific), 30,000 T cells and 15,000 target cells were co-cultured in the presence of 1, 0.2, 0.04, or 0 nM scDb in a total volume of 100 µl and E:T of 2:1. The co-culture was incubated for 20 h at 37 °C with 5% $CO_2$. Cell culture supernatant was removed and frozen for downstream analysis. The remaining cells were assayed for viability using the SteadyGlo assay (Promega) following the manufacturer's instructions. Luminescence was detected using a Synergy H1 Multi-Mode Reader (BioTek). Percent cytotoxicity was determined by the following formula: 1 − (Test Condition Luminescence/No scDb Control Condition Luminescence) × 100%. Frozen supernatants were thawed for IFNγ measurement using the Human IFNγ Quantikine ELISA kit (DIF50C R&D). Absorbance was measured at 450 and 540 nm using a Synergy H1 Multi-Mode Reader (BioTek). Reported $A_{450}$ values include an $A_{540}$ correction (subtract $A_{540}$ from $A_{450}$ for each well). All experiments were performed in triplicate.

### Molecular dynamics simulations of the WT, G12V, and G12V-to-WT KRAS-pHLA complexes

The CHARMM36m force field (March 2019 revision) was used for all systems[80] and simulations were performed with GROMACS 2022[81]. The TIP3P water model was used for the solvent. Simulations of the $KRAS^{WT}$-pHLA complex contained ~67,600 atoms and were performed in a rhombic dodecahedron box with a distance of 12 Å between the solute and the box, such that the distance between any two periodic images of the complex was 2.40 nm. Simulations of the $KRAS^{G12V}$-pHLA complex contained ~69,300 atoms and the same box specifications. The systems containing the complex in which the $KRAS^{G12V}$ peptide was reverted to the wild-type sequence also had ~69,300 atoms and the same box specifications. To perform the reversion of $KRAS^{G12V}$ to $KRAS^{WT}$, residue $Val12^{G12V}$ was mutated to $Gly12^{WT}$ using UCSF Chimera 1.15[82] via the *Tools -> Structure Editing -> Rotamers* dialog box. All

simulations were performed with a salt concentration of 0.15 M NaCl. In all simulations, the temperature was maintained at 310 K using V-rescale, a temperature coupling algorithm using velocity rescaling with a stochastic term, with a time constant of 0.1 ps. The pressure was isotropically maintained at 1 atm using the Parrinello-Rahman barostat with a time constant of 2 ps and a compressibility of $4.5^{10-5}$ $bar^{-1}$. Long-range electrostatic interactions were modeled using Smooth Particle Mesh Ewald (SPME) electrostatics, and non-bonded interactions were modeled with the Verlet cutoff scheme. The cutoff distance for short-range electrostatic and van der Waals interactions was set to 1.2 nm. A timestep of 2 fs was used, with atomic coordinates saved every 10 ps, and bonds containing hydrogen atoms were constrained using the LINCS algorithm[83]. Before all simulations, structures were energy minimized with the steepest descent algorithm, with an initial step size of 0.01 nm and a tolerance of 10 kJ/mol nm. Positional restraints of 1000 kJ/mol $nm^2$ on all heavy protein atoms were enforced during both the 2 ns NVT equilibration and 3 ns NPT equilibration for each system, and production MD simulations were conducted with positional restraints turned off. Three replicate production simulations were run for each system, such that each system was sampled for an aggregate time of 1.02 µs.

### Data analysis

Graphs were generated using GraphPad Prism 8 or GraphPad Prism 9. Statistical testing (ANOVA, multiple comparisons) were performed using GraphPad Prism 9. Molecular simulation data was processed using GROMACS 2022 binary programs *trjconv, trjcat,* and *rmsf.* The RMSF data obtained from the aggregate sampling of each system was plotted using Matplotlib[84], using the *matplotlib.pyplot* function (where *matplotlib.pyplot* was imported as *plt*) in an interactive Python3 Jupyter notebook.

### Reporting summary

Further information on research design is available in the Nature Portfolio Reporting Summary linked to this article.

## Data availability

The V2-IgG/$KRAS^{G12V}$-HLA-A*03:01 ternary complex data generated in this study have been deposited in the wwPBD database under accession code 7STF and Electron Microscopy Data Bank under accession code EMD-25427. The $KRAS^{WT}$-HLA-A*03:01 complex data generated in this study have been deposited in the wwPDB database under accession code 8DVG. The PDB ID 6O9B was used as the search model for the $KRAS^{WT}$-HLA-A*03:01 structure determination. The coordinates of the $KRAS^{WT}$-HLA-A*03:01 monomer (determined in this paper) PDB ID 8DVG and PDB entry 7KGU (2Q1-Fab) was used as a search model for the V2-IgG/$KRAS^{G12V}$-HLA-A*03:01 complex. Source data are provided with this paper. The SDS-PAGE gels, SPR sensorgrams, co-culture experiments and DSF data generated in this study are provided in the Supplementary Information/Source Data file. Source data are provided with this paper.

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

## Acknowledgements

The plasmids for expression of HLA-A*03:01 and β2-microglobulin were a gift from the NIH Tetramer Core Facility (Emory University, Atlanta, CA). Some illustrations were created with BioRender.com. We acknowledge the use of the computational resources from Walter and Eliza Hall Institute for cryo-EM data processing. The authors would like to thank Alisa Glukhova and Nicholas Kirk for invaluable discussions and suggestions about cryo-EM data processing. This work was supported by The Virginia and D.K. Ludwig Fund for Cancer Research; Lustgarten Foundation for Pancreatic Cancer Research; The Commonwealth Fund; The Bloomberg~Kimmel Institute for Cancer Immunotherapy; Bloomberg Philanthropies; NIH Cancer Center Support Grant P30 CA006973. J.D., B.J.M., A.H.P., & S.R.D. were supported by NIH Grant T32 GM73009. S.P. was supported by NCI K08CA270403, Leukemia Lymphoma Society Translational Research Program, American Society of Hematology Scholar Award, and Swim Across America Translational Cancer Research Award. M.F.K. was supported by NIH Grant T32 AR048522, The Jerome L. Greene Foundation Scholar Award, and The Cupid Foundation Discovery Award. C.B. was supported by NCI Grant R37 CA230400. Work at the AMX (17-ID-1) and FMX (17-ID-2) beamlines was supported by the NIH, the National Institute of General Medical Sciences (P41GM111244), the DOE Office of Biological and Environmental Research (KP1605010), and the National Synchrotron Light Source II at Brookhaven National Laboratory is supported by the DOE Office of Basic Energy Sciences under contract number DE-SC0012704 (KC0401040). The authors acknowledge the use of the Biacore Molecular Interaction Shared Resource (BMISR) at Georgetown University, which is supported by the National Institutes of Health (P30CA51008). Biorender was used for illustrations (Fig. 8a).

## Author contributions

Conceptualization: K.M.W., S.R.D., M.S.M., J.D., N.P., K.W.K., B.V., S.Z., S.B.G; Methodology: K.M.W., S.R.D., M.S.M., P.A.A., X.Z., Z.Y., M.C., J.D., M.S.H., E.H.H., B.J.M., A.H.P.; Investigation: K.M.W., S.R.D., M.S.M., P.A.A., X.Z., Z.Y., M.C., W.S.; Analysis and interpretation of data: K.M.W., S.R.D., M.S.M., P.A.A., M.C., J.D., M.S.H., E.H.H., B.J.M., A.H.P., S.P., M.F.K., D.M.P., C.B., N.P., K.W.K., B.V., S.Z., S.B.G; Writing—Original Draft: K.M.W., S.R.D., M.S.M.; Writing—review & editing: D.M.P., K.W.K., B.V., S.Z., S.B.G; Supervision: D.M.P., C.B., N.P., K.W.K., B.V., S.Z., S.B.G.

## Competing interests

The Johns Hopkins University has filed patent applications related to technologies described in this paper on which E.H.H., K.M.W., J.D., B.J.M., N.P., K.W.K., B.V., S.B.G. and S.Z. are listed as inventors: HLA-restricted epitopes encoded by somatically mutated genes (US20180086832A1), MANAbodies and methods of using (US20200079854A1), MANAbodies targeting tumor antigens and methods of using (PCT/US2020/065617). B.V., K.W.K. and N.P. are founders of Thrive Earlier Detection, an Exact Sciences Company. K.W.K. and N.P. are consultants to Thrive Earlier Detection. B.V., K.W.K., N.P. and S.Z. hold equity in Exact Sciences. B.V., K.W.K., N.P. and S.Z. are founders of, hold equity in, and serve as consultants to ManaT Bio. B.V., K.W.K., N.P. and S.Z. are founders of, hold equity in, and serve as consultants to Personal Genome Diagnostics. K.W.K., B.V., S.Z. and N.P. are consultants to and hold equity in NeoPhore. K.W.K. and N.P. are consultants to and own equity in Haystack Oncology. K.W.K., B.V., and N.P. hold equity in and are consultants to CAGE Pharma. B.V. is a consultant to and holds equity in Catalio Capital Management and may be a consultant to and hold equity in Haystack Oncology. S.Z. has a research agreement with BioMed Valley Discoveries, Inc. C.B. is a consultant to Depuy-Synthes, Galectin Therapeutics, Haystack Oncology, Privo Technologies and Bionaut Labs. C.B. is a co-founder of OrisDx. Patent applications on the work described in this abstract may be filed by Johns Hopkins University. The terms of all these arrangements are being managed by Johns Hopkins University in accordance with its conflict of interest policies. S.B.G. is a founder and holds equity in AMS, LLC.. M.F.K. received consulting fees from Argenx. D.M.P. reports grant and patent royalties through institution from BMS, grant from Compugen, stock from Trieza Therapeutics and Dracen Pharmaceuticals, and founder equity from Potenza; being a consultant for Aduro Biotech, Amgen, Astra Zeneca (Medimmune/Amplimmune), Bayer, DNAtrix, Dynavax Technologies Corporation, Ervaxx, FLX Bio, Rock Springs Capital, Janssen, Merck, Tizona, and Immunomic- Therapeutics; being on the scientific advisory board of Five Prime Therapeutics, Camden Nexus II, WindMil; being on the board of directors for Dracen Pharmaceuticals. The remaining authors declare no competing interests.

## Additional information

[1]Department of Biophysics and Biophysical Chemistry, The Johns Hopkins School of Medicine, Baltimore, MD 21205, USA. [2]Howard Hughes Medical Institute, Chevy Chase, MD 20815, USA. [3]Bloomberg-Kimmel Institute for Cancer Immunotherapy, Sidney Kimmel Comprehensive Cancer Center, Baltimore, MD 21287, USA. [4]Ludwig Center, Sidney Kimmel Comprehensive Cancer Center, Johns Hopkins University School of Medicine, Baltimore, MD 21287, USA. [5]Lustgarten Pancreatic Cancer Research Laboratory, Sidney Kimmel Comprehensive Cancer Center, Johns Hopkins University School of Medicine, Baltimore, MD 21287, USA. [6]Janelia Research Campus, HHMI,19700 Helix Drive, Ashburn, VA 20147, USA. [7]Energy & Photon Sciences Directorate, Brookhaven National Laboratory, Upton, NY 11973, USA. [8]Case Center for Synchrotron Biosciences, Case Western Reserve University, Cleveland, OH 44106, USA. [9]Department of Biomedical Engineering, Johns Hopkins University, Baltimore, MD 21218, USA. [10]Department of Oncology, Johns Hopkins University School of Medicine, Baltimore, MD 21287, USA. [11]Division of Hematologic Malignancies and Bone Marrow Transplantation, Johns Hopkins University School of Medicine, Baltimore, MD 21287, USA. [12]Division of Rheumatology, Department of Medicine, Johns Hopkins University School of Medicine, Baltimore, MD 21224, USA. [13]Department of Neurosurgery, Johns Hopkins University School of Medicine, Baltimore, MD 21205, USA. [14]Department of Pathology, Johns Hopkins University School of Medicine, Baltimore, MD 21205, USA. [15]Department of Medicine, Johns Hopkins University School of Medicine, Baltimore, MD 21205, USA. [16]Present address: Discovery Chemistry, Protein and Structural Chemistry, Merck & Co, Inc, West Point, PA 19846, USA. [17]Present address: Walter and Eliza Hall Institute, Parkville, VIC 3052, Australia. [18]Present address: Novartis Institutes for BioMedical Research, 250 Massachusetts Ave, Cambridge, MA 02139, USA. [19]These authors contributed equally: Katharine M. Wright, Sarah R. DiNapoli, Michelle S. Miller. ✉e-mail: sbzhou@jhmi.edu; Sandra.gabelli@merck.com

