## [Peer Review File · Nature Communications]

REVIEWER COMMENTS

Reviewer #1 (Remarks to the Author):

In this manuscript, the authors have presented the structure of the V2 TCRm in complex with KRASG12VHLA-A*03:01 and suggest that hydrophobic interactions and an induced conformational change dictate V2 specificity for the KRAS G12V peptide. The authors also solved the structure of the wild-type G12 peptide bound to HLA-A*03:01. Based on the structural and mutational analysis, the authors suggest a model for discrimination between the mutant and the wild-type peptides solely on hydrophobic interactions.

The work presented here is an extension of the previous report by another group on the structural analysis of the KRASG12D-HLA-C*08:02 pHLA in complex with patient derived-TCRs. Authors suggest that the mechanism for G12V mutant is different from G12D mutant.

The manuscript does provide new insights and would be of interest to researchers working in this area. The structural data is not very well presented, and I have listed below some of my concerns:

PDB validation report provided for the crystal structure is not the one that needs to be submitted for manuscript review. The one submitted with the manuscript clearly says "Not for Manuscript Review."

Rwork/Rfree in the highest resolution shell is very close to the overall Rwork/Rfree values. This indicates that either % of reflections in the test set is too small or there is some typo here.

The particle count in the final reconstruction in the cryo-EM table shows 367,584 particles, whereas, in the workflow, the class that went into the final reconstruction shows 116,685 particles. Please clarify.

What was the regularization parameter (T or tau fudge factor) used in the 3d classification step?

For the reconstruction, it would be good if the authors could provide an Euler angle distribution.

The reported Ramachandran favorable statistic is a bit low at 81.12% for their refinement statistics. Also, the authors do not provide the molprobtity score or the clashscore.

Even though the authors have provided R_{meas} and R_{pim} values, the overall R-merge of the crystallography data is 36%, much higher than generally expected for a good dataset (despite the higher redundancy).

The last sentence of the abstract has a typo: exclusively.

Reviewer #2 (Remarks to the Author):

I enjoyed reviewing this manuscript. It is interesting and potentially impactful on multiple fronts. My particular fortes are in structural biology, as well as a decent understanding of the RAS pathway - I will discuss the work broadly, but my focus (and much of the value of my review) will be in these two areas.

Potential Impact:

This work stands to be impactful on several fronts. Specificity is indeed a major challenge for cancer therapies, and mutation associated neoantigens (MANAS) are one possibility of addressing this challenge. With affinity being a key factor in MANA antibody efficacy, the insights into how these antibodies work (and how they may be improved) are certainly valuable. Importantly, the work focuses on KRAS - an extremely high priority cancer target. The work also represents a relatively successful navigation of a particularly challenging protein production and structure/function study; the strategies reported herein may be applied generally to other MANA-antibody studies.

The key results:

The work centers on the structure/function studies of a single-chain diabody (scDb) ("V2") that selectively recognizes processed G12V mutant KRAS peptides presented by HLA-A*03:01. The authors have structurally and functionally characterized the binding determinants of V2 (and selected mutants) to mutant KRAS presented fragments, and have hypothesized how these binding determinants differ from those available with the equivalent WT KRAS fragment, thus leading to specificity. Key advances in this work include: a cryo-EM structure of the V2-IgG/KRASG12V-pHLA complex (evidently the first use of Cryo-EM for such a complex), the crystal structure of crystal structure of KRASWT-HLA-A*03:01, and the functional characterization of 18 single amino acid variants of V2.

Critical concern:

The structural comparison between WT and mutant RAS peptides presented by HLA*03:01 requires more scrutiny/validation. The authors note that they were unable to crystallize the KRAS G12V-pHLA alone, which would typically be the ideal comparison. However, the authors are also comparing a crystal structure with a cryo-EM structure, and that comes with added liabilities. We should be concerned that experimental artifacts, namely crystal contacts, play a significant role in the observed conformation of the bound WT KRAS peptide, which would critically impact several of the hypotheses presented regarding selectivity. Indeed the authors note that the WTKRAS-pHLA crystallized readily, but the mutant did not - suggesting that this region is potentially involved in a crystal contact. To allay this critical concern:

- 1) Please check the crystal structure for crystal contacts in the vicinity of the WT KRAS peptide to determine the potential extent of interference.
- 2) If there is reason for concern, perform rigorous molecular dynamics simulations of the system in the absence of the crystal contact. If the conformation of the bound WT KRAS peptide remains stable, then perhaps the observed conformation (and hypotheses derived therefrom) may be regarded as valid. If not, reevaluation of the hypotheses may be necessary.
- 3) Potentially helpful: molecular dynamics simulations of the KRASG12V-pHLA with the V2 IgG removed - both mutant (as is), and with G12V mutated back to WT. Results could add confidence regarding any suggestion of induced fit / conformational differences between the KRAS peptides.

Ironically, if crystal contacts prove to be a concern, this may be an opportunity to further highlight the value of Cryo-EM structures. The authors mention that this is the first cryo-EM structure of an antibody-pHLA structure but do not fully capitalize on why that might be important. It is significant that they have, for the first time, demonstrated that Cryo-EM can be used in such systems - despite their inherent flexibility - and still arrive at quality high-resolution data for the key areas (peptide and HLA-antibody interface). Moreover, cryo-EM circumvents the need for crystallization, and avoids potential crystallization artifacts, which can be misleading...

Other critical points to address:

- The crystal structure deposition report is preliminary and clearly marked "Not For Manuscript Review". Please finalize and resubmit.
- The cryo-EM structure deposition report indicates a very poor clash score: there are 449 close contacts, many of them severe. Please confirm and comment on the location of these clashes relative to the critical areas of the structure.
- Abstract: "Here, we describe the first cryo-EM structure of any antibody-antigen complex". Definitely not true as written (there are many antibody-antigen complexes available). It may be the first reported cryo-EM structure of a (TCRm) antibody-MANA complex though. Please confirm that this was the intended statement, and that it is indeed true (and perhaps confirm and acknowledge that other

antibody-MANA complexes have been solved by crystallography, including reference 29 and well as others (eg <https://www.rcsb.org/structure/7BBG>)).

- Results: "...pointing towards a2 of HLA-A*03:01 (Fig. 3D, E, S4)." - there is no figure 3E.

Less critical clarifications:

- Results: "Indirect" and "allosteric" KRAS inhibitors should likely be "covalent" and "non-covalent", all of these inhibitors target the same induced pocket under switch 2 and only significantly differ in reversibility of binding, as far as I am aware.

- Results: Is the author sure that "identifying selective allosteric inhibitors" was the main challenge, or was it more about achieving potent inhibition of what was believed for decades to be an undruggable target? I have not read all of the given references, but my impression was that it was the latter.

- 149 heavy-atom to heavy-atom contacts, I presume? I would suggest moving the contact definition from the figure legend into the main body.

- "The KRASWT peptide buries a larger surface in the HLA-A*03:01 binding pocket compared to KRASG12V" - at a ~2% difference, 13 square angstroms is fairly negligible for buried surface area, particularly if crystal contacts turn out to be an issue. In any case, I would suggest retitling the section to focus on the hypothesized conformational differences, assuming those turn out to be valid.

- "Whereas residues Val8WT, Val9WT and Ala11WT formed hydrogen bonds to HLA-A*03:01, residues Val8G12V-Gly13G12V were only involved in hydrophobic interactions with no direct stabilization of the backbone." This seems odd given the similar placements of V8, V9 in the overlay in Figure 5C. Please check that those interactions are indeed different in the two structures.

- If the above is true, and there are indeed significant differences in hydrogen bonding of the N-terminal portions of the peptide, shouldn't that have shown up in the DSF experiments? As presented, the DSF "suggests that the KRASWT-HLAA*03:01 and KRASG12V-HLA-A*03:01 have similar structures".

- "In addition, the comparable binding to KRASG12WT-HLA-A*03:01, KRASG12V-HLA-A*11:01 and KRASG12WT-HLA-A*11:01 at high scDb concentrations of V2 scDb suggests a baseline degree of V2 binding to HLA-A3 superfamily members." Are you saying that at high concentrations of antibody you can get saturation in all three systems? I'm not sure this would lead to the conclusion suggested.

Further information I was left seeking as an interested reader:

- What is the lowest limit of density that a mutant cancer peptide has been successfully therapeutically targeted?

- Where is the S1 movie showing IgG flexibility - I could not find it in the reviewer package.

- Comment on disparity between Elisa results and SPR results?

- A sentence or two describing HLA variability in humans?

Reviewer #3 (Remarks to the Author):

Review for:

Hydrophobic interactions dominate the recognition of a KRAS G12V neoantigen

Katharine M. Wright, Sarah R. DiNapoli, Michelle S. Miller, P. Aitana Azurmendi, Xiaowei Zhao, Zhiheng Yu, WuXian Shi, Jacqueline Douglass, Michael S. Hwang, Emily Han-Chung Hsiue, Brian J Mog, Alexander H Pearlman, Suman Paul, Maximilian F Konig, Drew M Pardoll, Chetan Bettegowda, Nickolas Papadopoulos, Kenneth W Kinzler, Bert Vogelstein, Shibin Zhou, Sandra B. Gabelli

In this manuscript, Wright and colleagues investigated the presentation of the MANAs peptide (namely KRAS-G12V) by HLA-A*03:01 and its recognition by V2 TCR mimic antibody via Cryo-EM and crystallographic studies. The structural studies revealed hydrophobic interactions at the complex interface. However, the presented Cryo-EM data was at $\sim 3.3\text{\AA}$, which is a low resolution for investigating the detailed interaction at the interface between the peptide and V2 TCRm. The authors next generated various V2 variants based on the structural data, and identified two variants (F53W & V104R) to bind more tightly to KRASG12V-HLA, but without any increase in their sensitivity to the KRAS-G12V peptide in the cellular assays. Finally, Wright et al. investigated cross-reactivities of V2 to KRAS-G12V presented by HLA-A*11:01, and found its weaker binding to the V2.

Major points

- Fig 1, while the cryo-EM studies were performed by V2-Fab complex with HLA, Figure, 1A-B was for the full-length complex. It will be more informative to include similar panels for the Fab-HLA complex too.
- The CH and CL of the fab fragment seem very flexible in the structure (without a clear map to build in), which may hinder the correct fitting of their models in the map. Further, the data resolution (3.3A) is a bit low to investigate the basis of the interaction between the V2 and the peptide (Fig 1-2). The orientation of the V2/peptide/map in Fig 3S is not the best to judge here. For example, in Fig S3D, the CDR loop did not fit well in the shown map. I'm not an expert in the Cryo-EM, but I think further purification steps, more particles and/or refinement might assist to improve the resolution/map.
- The authors mentioned that CDRs of the Fab made 26 contacts with the peptide (Fig 3C), but they did not show/list the contacts anywhere. Figure 3 only showed the overall/zoomed docking of the loop atop the peptide without demonstrating the contacted residues and the types of contacts. A table of contacts will be helpful here. Ligplot in Fig S4 is not sufficient. A zoomed figure panel (may be based on Fig S4B) showing detailed contacts will be necessary to explain the text on page 6.

- Figure 3E was mentioned on page 6, but couldn't find it.
- The crystal structure of HLA-A*03:01 with KRAS (WT) is well analysed (Fig 4) and the quality is good.
- Unfortunately, the crystallization trials of HLA-A*03:01 with KRAS (G12V) were unsuccessful in the comparison to the complex structure, however, the authors did not explain whether they tried Cryo-EM here.
- The authors assume that the specificity of the V2 binding is based upon the induced fit of the loose, hydrophobic cage of V2-CDR with G12V of the peptide, yet they don't have any structural data of the free V2 fab to confirm this hypothesis.

Reviewer #4 (Remarks to the Author):

The manuscript by Wright et al. described a progress in understanding interactions between a KRAS G12V neoantigen-bound HLA and a V2 antibody. The authors conducted cryo-EM structural determination of the complex of pHLA-V2 and also determined the crystal structure of HLA bound to the wild type KRAS peptide. The comparison of the two structures indicated induced fit of the KRAS G12V neoantigen-HLA to V2 and there were substantial hydrophobic interactions involved in the binding of V2 to the KRAS G12V neoantigen-HLA. Based on the determined structures, a large V2 library was created and selected to identify more potent V2 variants. Results were mixing. Some showed high potency but not achieving both strong potency and selectivity. Although their mutation results were not as impressive as the structural determination part, the work itself is very important for others who may follow on this research. The manuscript is well written and leaves very little for criticism. There is only one comment from this review that the author may consider for their revision of the draft.

1. Cysteine in many ways behave like a hydrophobic residue. Its size is also very similar to valine. It is not a surprise that a developed antibody doesn't differentiate between G12V and G12C neoantigens.

REVIEWER COMMENTS

Reviewer #1 (Remarks to the Author):

PDB validation report provided for the crystal structure is not the one that needs to be submitted for manuscript review. The one submitted with the manuscript clearly says "Not for Manuscript Review."

We apologise for submitting the incorrect file. The correct validation report has now been included.

Even though the authors have provided Rmeas and Rpim values, the overall R-merge of the crystallography data is 36%, much higher than generally expected for a good dataset (despite the higher redundancy).

We thank the reviewer for bringing this to our attention. We agree that the R-merge is quite high, but given the low Rpim, we reasoned this is primarily a result of the high redundancy. To address the reviewer's concerns, we have reprocessed the data by reducing the total number of frames, which has reduced the R-merge to 24%. The CC1/2 in the highest resolution shell (76%) and the overall Rpim (7%) are well within acceptable limits.

Rwork/Rfree in the highest resolution shell is very close to the overall Rwork/Rfree values. This indicates that either % of reflections in the test set is too small or there is some typo here.

The test set has 5% of the total reflections. We have updated the table with the values from the reprocessed and re-refined structure.

The particle count in the final reconstruction in the cryo-EM table shows 367,584 particles, whereas, in the workflow, the class that went into the final reconstruction shows 116,685 particles. Please clarify.

Our apologies, the incorrect number was transferred to the table, but was correct in the workflow (Fig. S2). This has been corrected in the table to 116,685 particles.

What was the regularization parameter (T or tau fudge factor) used in the 3d classification step?

We have added the following sentence to the methods and the figure legend Fig. S2, 'A value of T=4 was used for both 3D classification steps'.

For the reconstruction, it would be good if the authors could provide an Euler angle distribution.

We have included the Euler angle distribution as a new supplementary figure (Fig S3).

The reported Ramachandran favorable statistic is a bit low at 81.12% for their refinement statistics. Also, the authors do not provide the molprobity score or the clashscore.

We thank the reviewer for bringing this to our attention. The molprobity score has been added to Table 1.

The last sentence of the abstract has a typo: exclusively.

This has been corrected.

Reviewer #2 (Remarks to the Author):

1) Please check the crystal structure for crystal contacts in the vicinity of the WT KRAS peptide to determine the potential extent of interference.

We thank the reviewer for allowing us to address this. Below is a list and image of crystal contacts and other observations upon inspection of the KRAS^{WT}-HLA-A*03:01 crystal structure:

- 1- Arg65 guanidinium group from symmetry mate HLA-A*03:01 inserts between the KRAS^{WT} peptide and alpha2 helix of HLA-A*03:01.
- 2- Carbonyl of Gly6 of KRAS^{WT} peptide is at hydrogen bonding (H-bonding) distance to a water that is at H-bonding distance to the carbonyl of HLA-A*03:01 Arg65 symmetry mate.
- 3- Comparing the mutant and wild-type peptides of the two structures, the mutant G12V conformation could not be accommodated with the crystal packing (alanine flip and valine/valine clash).

Upon these observations and suggestion from the reviewer, we carried out molecular dynamics experiments (see page 9 of the main text and supplemental figure Fig. S7 as well as responses below).

2) *If there is reason for concern, perform rigorous molecular dynamics simulations of the system in the absence of the crystal contact. If the conformation of the bound WT KRAS peptide remains stable, then perhaps the observed conformation (and hypotheses derived therefrom) may be regarded as valid. If not, re-evaluation of the hypotheses may be necessary.*

We thank the reviewer for this suggestion. We have carried out molecular dynamics simulations of the KRAS^{WT}-pHLA system for an aggregate sampling time of 1.02 μ s. In these simulations, the distance between the complex and the edge of the solvent box was set to 1.2 nm, such that any periodic images of the KRAS^{WT}-pHLA complex would be 2.4 nm apart during the simulation. Quantifying the root mean square fluctuation (RMSF) of the KRAS^{WT} peptide, we observe that all residues, except for the N-terminal Gly7^{WT}, exhibit a fluctuation of less than 1.5 Å. This suggests that the conformation of the bound KRAS^{WT} peptide remains stable relative to its originally observed pose, and corroborates the observations and hypotheses obtained from the crystal structure. A short paragraph describing this

observation has been added to page 9 of the main text, and a supplemental figure (Fig. S7) showing this data has been added.

3) *Potentially helpful: molecular dynamics simulations of the KRAS^{G12V}-pHLA with the V2 IgG removed - both mutant (as is), and with G12V mutated back to WT. Results could add confidence regarding any suggestion of induced fit / conformational differences between the KRAS peptides.*

We thank the reviewer for this helpful suggestion. We have carried out molecular dynamics simulations of the KRAS^{G12V}-pHLA system with the V2 IgG removed, and simulations in which KRAS^{G12V} was mutated back to KRAS^{WT}. As in the case of the KRAS^{WT}-pHLA system above, both simulations have been conducted for an aggregate sampling time of 1.02 μ s, and we have quantified the RMSF of both peptides. We observe that KRAS^{G12V} exhibits the largest fluctuations, while the system in which KRAS^{G12V} was mutated back to WT exhibits fluctuations greater than the original KRAS^{WT} but lower than KRAS^{G12V}. We posit the intermediate fluctuations of the reverted KRAS^{G12V} to WT system to be a 'memory' effect resulting from its initial conditions, in which it was KRAS^{G12V} and was making hydrophobic contacts with the V2 IgG. The reversion to KRAS^{WT} likely tempered the extent of the fluctuations observed during simulation. We can interpret the RMSF results to have two principal implications. Firstly, the removal of the V2 IgG in the simulation, representing a removal of favorable hydrophobic contacts with the KRAS^{G12V}, is also observed to increase the fluctuations of the peptide residues, indicating its influence in inducing a conformational change of the peptide. Secondly, we posit that the increased fluctuations of KRAS^{G12V} relative to the original KRAS^{WT} may allow it to adopt additional conformational states (i.e., increase its conformational entropy) in a way that facilitates its induced conformational pose when interacting with the V2 IgG, and thereby confer its recognition specificity. We have added a short paragraph to pages 10 – 11 of the main text that summarize the points made here, as well as a supplemental figure (Fig. S7).

Ironically, if crystal contacts prove to be a concern, this may be an opportunity to further highlight the value of Cryo-EM structures. The authors mention that this is the first cryo-EM structure of an antibody-pHLA structure but do not fully capitalize on why that might be important. It is significant that they have, for the first time, demonstrated that Cryo-EM can be used in such systems - despite their inherent flexibility - and still arrive at quality high-resolution data for the key areas (peptide and HLA-antibody interface). Moreover, cryo-EM circumvents the need for crystallization, and avoids potential crystallization artifacts, which can be misleading...

We thank the reviewer for the opportunity to highlight the significance of the cryoEM structure in the manuscript.

The following paragraph has been added to the discussion:

'This report includes the first cryo-EM structure of an antibody fragment binding a MANA pHLA target, and the first structures of the KRAS^{WT/G12V}₇₋₁₆ peptides presented by HLA-A*03:01 with or without an antibody in complex. Despite the inherent flexibility and dynamic structures of full-length IgG's, cryo-EM allows for high-resolution visualisation of a Fab-HLA interaction without the limitation of crystallization artifacts that could influence protein conformation. While the KRAS^{WT}-HLA-A*03:01 peptide is involved in crystal packing, the molecular dynamics simulations conducted in this work support the observed KRAS^{WT} peptide conformation and the conformational flexibility of the KRAS^{G12V} peptide upon V2-Fab binding. Moreover, such techniques and approaches can be applied to systems where conformational flexibility may be important for selectivity, or in which crystallization artifacts may complicate structural interpretation.'

Other critical points to address:

- The crystal structure deposition report is preliminary and clearly marked "Not For Manuscript Review". Please finalize and resubmit.

We apologise for uploading the incorrect report. The correct one has now been supplied.

- The cryo-EM structure deposition report indicates a very poor clash score: there are 449 close contacts, many of them severe. Please confirm and comment on the location of these clashes relative to the critical areas of the structure.

We thank the reviewer for allowing us to address the clashes. Upon further investigation of the PDB validation report and after the removal of the hydrogen-hydrogen clashes, most of the remaining clashes are outside the 'masked' region (Masked region: Chain A – HLA-A3 residues 2-180; Chain H – Variable heavy chain residues 1-116; Chain L – Variable light chain residues 1-106; Chain C – KRAS peptide). Therefore, the lack of resolution in the area limits the rebuilding.

- Abstract: "Here, we describe the first cryo-EM structure of any antibody-antigen complex". Definitely not true as written (there are many antibody-antigen complexes available). It may be the first reported cryo-EM structure of a (TCRm) antibody-MANA complex though. Please confirm that this was the intended statement, and that it is indeed true (and perhaps confirm and acknowledge that other antibody-MANA complexes have been solved by crystallography, including reference 29 and well as others (eg <https://www.rcsb.org/structure/7BBG>)).

We thank the reviewer for bringing this to our attention.

The intended statement was meant to be "Here, we describe the first cryo-EM structure of any antibody-MANA pHLA complex." We had emphasized this a second time in the discussion with the following sentence, 'This report includes the first cryo-EM structure of an antibody fragment binding a pHLA target and the first structures of the KRAS^{WT/G12V}₇₋₁₆ peptides presented by HLA-A*03:01 with or without an antibody in complex.' We have confirmed this is true, and the sentence has been corrected in the abstract.

In regards to acknowledging other antibody-MANA complexes determined by crystallography, we had written the following as part of the introduction: 'The determination of the structures of MANA-targeting therapeutics could yield unique information about how TCRs and antibody-based immunotherapies recognize pHLA, and provide opportunities for their improvement (33,34). For example, structural analysis of the KRAS^{G12D}-HLA-C*08:02 pHLA in complex with patient derived-TCRs revealed that the G12D mutation is a critical anchor residue for peptide presentation, but is not directly involved in TCR recognition of the neoantigenic peptide (23). Others have compared affinity-enhanced TCRs and TCR mimic (TCRm) pHLA-targeting antibodies using crystal structures to understand the differences in binding affinity and specificity of agents with shared pHLA targets (34).' We believe these highlights other MANA-antibody structures determined by crystallography.

- Results: "...pointing towards $\alpha 2$ of HLA-A*03:01 (Fig. 3D, E, S4)." – there is no figure 3E.

We apologize for the confusion. The reference to Fig. 3E has been removed from the paper.

Less critical clarifications:

- Results: "Indirect" and "allosteric" KRAS inhibitors should likely be "covalent" and "non-covalent", all of these inhibitors target the same induced pocket under switch 2 and only significantly differ in reversibility of binding, as far as I am aware.

We thank the reviewer for pointing this out and acknowledge the reviewer is correct. The sentence has been changed to the following, 'Recently, groups have identified covalent inhibitors to KRAS^{G12C} and non-covalent inhibitors to KRAS^{G12D} that offer promise for small molecule targeting of this previously "undruggable" target.'

- Results: Is the author sure that “identifying selective allosteric inhibitors” was the main challenge, or was it more about achieving potent inhibition of what was believed for decades to be an undruggable target? I have not read all of the given references, but my impression was that it was the latter.

We thank the reviewer for allowing us to elaborate.

Following the confusion of the reviewer, we rewrote that sentence to highlight all the challenges encountered in targeting KRAS.

Many of the challenges documented in the literature include: the featureless structure of KRAS, inaccessibility of the GTP/GDP binding pocket, identification of additional binding pockets, and selectivity for the KRAS mutants over the wild-type KRAS protein.

For clarification, the sentence has been changed to the following,

‘For decades, targeting codon 12 mutant KRAS proteins with small molecule inhibitors was impeded by the inaccessibility of the GTP/GDP binding pocket, the featureless structure of KRAS, the lack of secondary binding pockets, and challenges in identifying selective inhibitors over the wild-type KRAS protein (5-10).’

- 149 heavy-atom to heavy-atom contacts, I presume? I would suggest moving the contact definition from the figure legend into the main body.

Yes, the 149 contacts refer to heavy-atom to heavy-atom contacts.

We thank the reviewer for the suggestion and have added the contact definition from the figure legend into the main body (page 6).

- “The KRAS^{WT} peptide buries a larger surface in the HLA-A*03:01 binding pocket compared to KRAS^{G12V}” – at a ~2% difference, 13 square angstroms is fairly negligible for buried surface area, particularly if crystal contacts turn out to be an issue. In any case, I would suggest retitling the section to focus on the hypothesized conformational differences, assuming those turn out to be valid.

We agree with the reviewer and renamed the section ‘The KRAS^{WT} peptide binds in the HLA-A*03:01 binding pocket.’

- “Whereas residues Val8^{WT}, Val9^{WT} and Ala11^{WT} formed hydrogen bonds to HLA-A*03:01, residues Val8^{G12V}-Gly13^{G12V} were only involved in hydrophobic interactions with no direct stabilization of the backbone.” This seems odd given the similar placements of V8, V9 in the overlay in Figure 5C. Please check that those interactions are indeed different in the two structures.

We appreciate the opportunity to better highlight the data. While the placement of Val8 and Val9 between the two structures overlay well despite the loss of hydrogen bonds, hydrophobic interactions could be compensating and keeping the peptide anchored. The two hydrogen bonds lost in the KRAS^{G12V}-V2 Fab structure when compared to the KRAS^{WT} structure were between the amino group of Val8 and Asp63 (HLA-A3) and the amino group of Val9 and Tyr99 (HLA-A3).

Below are images, centered on the N-terminus of the peptide, highlighting the placement of these residues within each structure, and the corresponding electron density map.

While the N-terminus of the peptides remain unchanged, there is a clear difference in rotamer of Asp63 and Tyr99 of the HLA-A3 between the structures that affects their positioning. Due to this difference, hydrogen bonds to the peptide could no longer be made.

CryoEM electron density map of KRAS^{G12V}-HLA-A3/V2-Fab

Xray crystallography electron density map of KRAS^{WT}-HLA-A3

- If the above is true, and there are indeed significant differences in hydrogen bonding of the N-terminal portions of the peptide, shouldn't that have shown up in the DSF experiments? As presented, the DSF "suggests that the KRAS^{WT}-HLA-A*03:01 and KRAS^{G12V}-HLA-A*03:01 have similar structures".

The DSF experiments were performed with the pHLA monomers in the absence of antibody. In the manuscript, since we were not able to crystallize the KRAS^{G12V}-HLA-A*03:01 monomer, we compared the differences between the KRAS^{WT}-HLA-A*03:01 and the KRAS^{G12V}-HLA-A*03:01 bound to V2-Fab. Based on the structural observance of a conformational change and additional molecular dynamics data, the changes in hydrogen bonding of the peptide to the HLA that we see could be due to binding of the antibody. However, in the absence of antibody, the interactions could be similar, hence the similar melting temperatures by DSF.

- "In addition, the comparable binding to KRAS^{G12V}-HLA-A*03:01, KRAS^{G12V}-HLA-A*11:01 and KRAS^{G12V}-HLA-A*11:01 at high scDb concentrations of V2 scDb suggests a baseline degree of V2 binding to HLA-A3 superfamily members." Are you saying that at high concentrations of antibody you can get saturation in all three systems? I'm not sure this would lead to the conclusion suggested.

We agree with the reviewer. Further inspection of the paragraph, we do not believe that the sentence adds content to the overall conclusions. We have removed the sentence.

Further information I was left seeking as an interested reader:

- What is the lowest limit of density that a mutant cancer peptide has been successfully therapeutically targeted?

We thank the reviewer for the further interest. We added a sentence specifying the low antigen density in the introduction: "These peptides are typically detected at single-digit to tens of copies per cell."

We have developed bispecific antibodies targeting several mutation-associated neoantigens (KRAS G12V and p53 R175H) where we determined the copies of mutant peptide per cell via mass spectrometry (Douglass Sci. Immunol. 2021, Hsiue Science 2022). In those reports, mutant peptides were present at 1-10 copies per cell in cancer cell lines with endogenous mutations. Others have developed modified T-cell receptor (TCR)-derived bispecific antibodies to target cancer testis antigens such as NY-ESO-1 and MAGE-A3 and reported typical antigen densities of 14-15 copies per cell (Liddy et al. Nature Medicine 2012). These ImmTAC bispecific antibodies had activity at lower antigen density as well (2-10 copies per cell). Others have used adoptive T cell therapy where transgenic T cell expressing MANA-specific TCRs were administered to patients. TCRs have reported sensitivity to even just 1 cognate antigen (Irvine Nature 2002), suggesting that an optimized immunotherapeutic format (transgenic TCRs, bispecific antibodies, CAR T cells) would be able to target cancer cells with only 10s of mutant peptide presented.

- Where is the S1 movie showing IgG flexibility – I could not find it in the reviewer package.

We apologize for the confusion.

The video has been added to the submission.

- Comment on disparity between Elisa results and SPR results?

We thank the reviewer for their suggestion. While affinity changes for V104N and V104R corresponded well between the monomer ELISA and SPR data, the patterns for F53W did not. One possible explanation is differences in antibody concentration, however the anti-CD3 ELISA absorbance for F53W is comparable to other scDBs tested (Fig. S10). Both V104R and V104N had 5-fold differences in affinity (higher and lower), while F53W had a 3-fold decrease in affinity. Estimating relative affinity via ELISA may not be sensitive enough to demonstrate smaller changes in affinity. We added a further comment in the results: "F53W^{L2} had lower relative binding on ELISA, but a higher affinity by SPR ($K_D = 10.6$ nM), indicating that affinity measurement by SPR may be more sensitive to smaller changes in affinity than ELISA (Fig. 6E; Table S3)."

- A sentence or two describing HLA variability in humans?

We thank the reviewer for their comment. Pearlman et al, Nat Cancer 2021 highlighted nicely the phenotype frequencies of the ten most common HLA alleles in the United States and common neoantigens across cancers. A summary sentence was added to the results page 15: 'In the United States, the 10 most common HLA alleles range in frequency from 16-42% of the population, with HLA-A*03:01 present at a frequency of 22% (1).'

Reviewer #3 (Remarks to the Author):

Major points

- Fig 1, while the cryo-EM studies were performed by V2-Fab complex with HLA, Figure, 1A-B was for

the full-length complex. It will be more informative to include similar panels for the Fab-HLA complex too.

We thank the reviewer for allowing us to clarify.

The cryo-EM studies were performed with the full-length V2-IgG (150 kDa) in complex with the KRAS^{G12V}-HLA-A*03:01. The studies were not performed with V2-Fab. Due to flexibility in the V2-IgG hinge region, only the V2 Fab-pHLA structure was resolved from the collected cryo-EM data. The first paragraph in the results section outlines the workflow of how we decided to use the full-length V2-IgG instead of the V2-Fab.

'Initial attempts at structure determination were performed using the V2 single-chain variable fragment (scFv) in complex with the KRAS^{G12V}-HLA-A*03:01 monomer. We could not get this protein expressed at high levels, despite several attempts, and so we switched to a full-length IgG format, grafting the V2 scFv into a full-length immunoglobulin G1 (IgG1) framework (V2-IgG). We performed pepsin-digestion and reduction of the V2-IgG into an antibody-fragment (V2-Fab') (Fig. S1), but crystallization of the V2-Fab'-pHLA complex was unsuccessful. We then attempted to use size exclusion chromatography (SEC) to purify the full-length V2-IgG in complex with the KRAS^{G12V}-HLA-A*03:01 monomer. Complex formation was confirmed by a shift in the SEC elution pattern (Fig. 1A, B). The high molecular weight of the V2-IgG/KRAS^{G12V}-pHLA complex (150 kDa + 2 x 46 kDa) seemed ideal for single particle cryo-electron microscopy (cryo-EM).'

- The CH and CL of the fab fragment seem very flexible in the structure (without a clear map to build in), which may hinder the correct fitting of their models in the map. Further, the data resolution (3.3A) is a bit low to investigate the basis of the interaction between the V2 and the peptide (Fig 1-2). The orientation of the V2/peptide/map in Fig 3S is not the best to judge here. For example, in Fig S3D, the CDR loop did not fit well in the shown map. I'm not an expert in the Cryo-EM, but I think further purification steps, more particles and/or refinement might assist to improve the resolution/map.

As it can be observed in Fig. 1B, the purity of the V2-IgG is very good. Further purification will not improve resolution of the map in this case. The resolution of the map or lack thereof most likely reflects the inherent flexibility of the complex and/or quality of the data and at > 3.1 Å not all the side chains will be observed or fit ideally. However, the resolution was sufficient to see detailed interactions at the HLA-peptide-V2 Fab interface.

- The authors mentioned that CDRs of the Fab made 26 contacts with the peptide (Fig 3C), but they did not show/list the contacts anywhere. Figure 3 only showed the overall/zoomed docking of the loop atop the peptide without demonstrating the contacted residues and the types of contacts. A table of contacts will be helpful here. Ligplot in Fig S4 is not sufficient. A zoomed figure panel (may be based on Fig S4B) showing detailed contacts will be necessary to explain the text on page 6.

We have added a list of the 26 contacts as a Supplemental Table S2 as analysed by Contacts in the CCP4 suite.

Regarding Figure S4 (now S5), the 26 contacts noted are between the KRAS^{G12V} peptide (blue) and the heavy chain CDRs (H1- magenta and H3- orange of the V2-Fab) and the light chains CDRs (L2- cyan). The LigPlot depicts hydrophobic contacts to the residue with a dashed arc regardless of if there are one or multiple hydrophobic contacts present. In this case, a zoomed-in image will not show any other information. The hydrogen bonds marked in Fig. S4A (now Fig. S5A) are between the KRAS^{G12V} peptide (blue) and HLA-A*03:01 (grey).

- Figure 3E was mentioned on page 6, but couldn't find it.

We apologize for the confusion. The reference to Fig. 3E has been removed from the paper.

- Unfortunately, the crystallization trials of HLA-A*03:01 with KRAS (G12V) were unsuccessful in the comparison to the complex structure, however, the authors did not explain whether they tried Cryo-EM here.

Due to the low-signal to noise ratio of cryo-EM images, higher molecular weight species are considered more robust for data collection and structure determination. The KRAS^{G12V}-HLA-A*03:01 monomer is 47 kDa and generally considered too small for cryo-EM. While there is a cryo-EM structure of streptavidin (52 kDa) to 3.2 Å, most of the cryo-EM structures in literature are of larger molecules with very few of smaller proteins. Due to this analysis, we deemed the possibility of success very low and cryoEM was not attempted with KRAS^{G12V}-HLA-A*03:01.

- The authors assume that the specificity of the V2 binding is based upon the induced fit of the loose, hydrophobic cage of V2-CDR with G12V of the peptide, yet they don't have any structural data of the free V2 fab to confirm this hypothesis.

We thank the reviewer for allowing us to address this.

The structures presented in this manuscript highlight the induced fit of the peptide upon binding to the V2-Fab. Since the induced fit we observe is in the conformation of the KRAS peptide, we do not believe that the structure of the V2-Fab alone would provide further information about binding, since it is known that the CDRs are quite flexible and change orientation.

Reviewer #4 (Remarks to the Author):

1. Cysteine in many ways behave like a hydrophobic residue. Its size is also very similar to valine. It is not a surprise that a developed antibody doesn't differentiate between G12V and G12C neoantigens.

We thank the reviewer for their comment. It is possible that the difference between G12V and G12C is not sufficient for an antibody to differentiate the peptide. We decided there was insufficient data to conclude that the V2 variants had an improved/desirable dual specificity for G12C and G12V. For example, the A106 variants had elevated IFN γ production in wild-type peptide pulsing conditions independent of peptide concentration (Fig. 7), suggesting non-specific binding to the HLA-A*03:01 pMHC or other cell surface antigens. Similarly, V104R had increased KRAS^{WT} binding when characterized by SPR (Fig. 6C). We have made several attempts to generate an isogenic KRAS^{WT} cell line via CRISPR editing to better assess the specificity the observed H358^{G12C} reactivity. While generating other G12 or G13 variants was possible, we were unable to identify viable KRAS^{WT/WT}, KRAS^{R^{WT}/-} or KRAS^{-/-} clones, likely due to requirements for mutant KRAS for cell line fitness. Previously, the specificity of the original V2 scDb was characterized both against the same G12C+ H358 cell line and against a single variant peptide library (Douglass Science Immunol 2021). Douglass et al. reported similar results with the H358 cell lines, and little reactivity to G12WT or G12C in the peptide pulsing co-cultures. This suggests that it is possible to differentiate between G12C and G12V with a sufficiently specific antibody.

REVIEWERS' COMMENTS

Reviewer #1 (Remarks to the Author):

In the revised manuscript, the authors have incorporated suggestions made during the initial review. I am satisfied with their responses and recommend the publication of this manuscript.

Reviewer #2 (Remarks to the Author):

The authors have addressed all of my concerns and queries.

In particular, they have sufficiently alleviated my main concern of crystallization artifacts significantly affecting their structural hypotheses by performing follow-up investigations and experiments to strengthen their structural hypotheses. With the follow-up information provided, I concur that that the crystal contacts likely have minimal effect on the KRAS peptide (and in particular the most relevant parts of it).

Incorporating the above further analyses along with addressing my other queries / incorporating my numerous other suggestions has strengthened the paper (imho). The work is consistent, and will be of significance to the field.

Reviewer #3 (Remarks to the Author):

My concerns have been reported

We appreciate the effort that the reviewers and the editorial team have put into making our manuscript into a compliant, improved version.

REVIEWERS' COMMENTS

Reviewer #1 (Remarks to the Author):

In the revised manuscript, the authors have incorporated suggestions made during the initial review. I am satisfied with their responses and recommend the publication of this manuscript.

We thank the reviewer.

Reviewer #2 (Remarks to the Author):

The authors have addressed all of my concerns and queries.

In particular, they have sufficiently alleviated my main concern of crystallization artifacts significantly affecting their structural hypotheses by performing follow-up investigations and experiments to strengthen their structural hypotheses. With the follow-up information provided, I concur that that the crystal contacts likely have minimal effect on the KRAS peptide (and in particular the most relevant parts of it).

Incorporating the above further analyses along with addressing my other queries / incorporating my numerous other suggestions has strengthened the paper (imho). The work is consistent, and will be of significance to the field.

We thank the reviewer for allowing us to strengthen our conclusions.

Reviewer #3 (Remarks to the Author):

My concerns have been reported.

We thank the reviewer.